# EFFICIENT SUBGRAPH GNNS BY LEARNING EFFECTIVE SELECTION POLICIES

**Beatrice Bevilacqua**
Purdue University
`bbevilac@purdue.edu`

**Moshe Eliasof**
University of Cambridge
`me532@cam.ac.uk`

**Eli Meirom**
NVIDIA Research
`emeirom@nvidia.com`

**Bruno Ribeiro**
Purdue University
`ribeiro@cs.purdue.edu`

**Haggai Maron**
Technion & NVIDIA Research
`hmaron@nvidia.com`

## ABSTRACT

Subgraph GNNs are provably expressive neural architectures that learn graph representations from sets of subgraphs. Unfortunately, their applicability is hampered by the computational complexity associated with performing message passing on many subgraphs. In this paper, we consider the problem of learning to select a small subset of the large set of possible subgraphs in a data-driven fashion. We first motivate the problem by proving that there are families of WL-indistinguishable graphs for which there exist efficient subgraph selection policies: small subsets of subgraphs that can already identify all the graphs within the family. We then propose a new approach, called POLICY-LEARN, that learns how to select subgraphs in an iterative manner. We prove that, unlike popular random policies and prior work addressing the same problem, our architecture is able to learn the efficient policies mentioned above. Our experimental results demonstrate that POLICY-LEARN outperforms existing baselines across a wide range of datasets.

## 1 INTRODUCTION

Subgraph GNNs (Zhang & Li, 2021; Cotta et al., 2021; Papp et al., 2021; Bevilacqua et al., 2022; Zhao et al., 2022; Papp & Wattenhofer, 2022; Frasca et al., 2022; Qian et al., 2022; Zhang et al., 2023) have recently emerged as a promising research direction to overcome the expressive-power limitations of Message Passing Neural Networks (MPNNs) (Morris et al., 2019; Xu et al., 2019). In essence, a Subgraph GNN first transforms an input graph into a bag of subgraphs, obtained according to a predefined generation policy. For instance, each subgraph might be generated by deleting exactly one node in the original graph, or, more generally, by marking exactly one node in the original graph, while leaving the connectivity unaltered (Papp & Wattenhofer, 2022). Then, it applies an equivariant architecture to process the bag of subgraphs, and aggregates the representations to obtain graph- or node-level predictions. The popularity of Subgraph GNNs can be attributed not only to their increased expressive power compared to MPNNs but also to their remarkable empirical performance, exemplified by their success on the ZINC molecular dataset (Frasca et al., 2022; Zhang et al., 2023).

Unfortunately, Subgraph GNNs are hampered by their computational cost, since they perform message-passing operations on all subgraphs within the bag. In existing subgraph generation policies, there are at least as many subgraphs in the bag as there are nodes in the graph ($n$), leading to a computational complexity of $\mathcal{O}(n^2 \cdot d)$ contrasted to just $\mathcal{O}(n \cdot d)$ of a standard MPNN, where $d$ represents the maximal node degree. This computational overhead becomes intractable in large graphs, preventing the applicability of Subgraph GNNs on commonly-used datasets. To address this limitation, previous works (Cotta et al., 2021; Bevilacqua et al., 2022; Zhao et al., 2022) have considered randomly sampling a small subset of subgraphs from the bag in every training epoch, showing that this approach not only significantly reduces the running time and computational cost, but also empirically retains good performance.

This raises the following question: *is it really necessary to consider all the subgraphs in the bag?* To answer this question, consider the two non-isomorphic graphs in Figure 1, which are indistinguishable

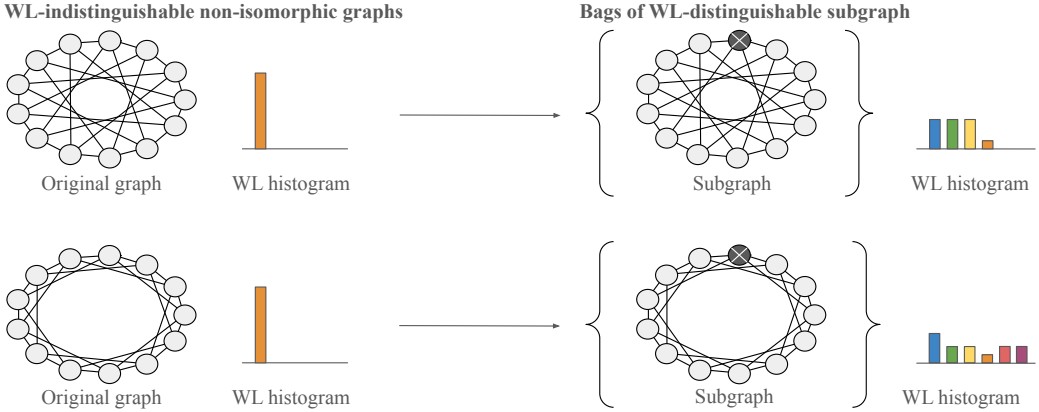

Figure 1: Sufficiency of small bags. Two non-isomorphic graphs (Circulant Skip Links graphs with 13 nodes and skip length 5 and 3, respectively) that can be distinguished using a bag containing only a single subgraph, generated by marking one node.

by MPNNs. As demonstrated by prior work (Cotta et al., 2021), encoding such graphs as bags of subgraphs, each obtained by marking a single node, makes them distinguishable. However, as can be seen from the Figure, the usage of all subgraphs is unnecessary, and distinguishability can be achieved by considering only one subgraph per graph. Thus, from the standpoint of expressive power, Subgraph GNNs can distinguish between these graphs using constant-sized bags, rather than the $n$-sized bags typically employed by existing methods.

The example in Figure 1 shows the potential of subgraph selection policies returning a small number of subgraphs. However, as all node-marked subgraphs in these two graphs are isomorphic, randomly selecting one subgraph in each graph is sufficient for disambiguation. In Section 4 we generalize this example and prove that there exist families of non-isomorphic graphs where a small number of *carefully chosen* subgraphs is sufficient and necessary for identifying otherwise WL-indistinguishable graphs. In these families, the probability that a random selection baseline returns the required subgraphs tends to zero, motivating the problem of learning to select subgraphs instead of randomly choosing them.

In this paper, we aim to learn subgraph selection policies, returning a significantly smaller number of subgraphs compared to the total available in the bag, in a task- and graph-adaptive fashion. To that end, we propose an architecture, dubbed POLICY-LEARN, composed of two Subgraph GNNs: (i) a *selection* network, and (ii) a downstream *prediction* network. The selection network learns a subgraph selection policy and iteratively constructs a bag of subgraphs. This bag is then passed to the prediction network to solve the downstream task of interest. The complete architecture is trained in an end-to-end fashion according to the downstream task's loss function, by leveraging the straight-through Gumbel-Softmax estimator (Jang et al., 2016; Maddison et al., 2017) to allow differentiable sampling from the selection policy.

As we shall see, our choice of using a Subgraph GNN as the selection network, as well as the sequential nature of our policy generation, provide additional expressive power compared to random selection and previous approaches (Qian et al., 2022) and allow us to learn a more diverse set of policies. Specifically, we prove that our framework can learn to select the subgraphs required for disambiguation in the families of graphs mentioned above, while previous approaches provably cannot do better than randomly selecting a subset of subgraphs.

Experimentally, we show that POLICY-LEARN is competitive with full-bag Subgraph GNNs, even when learning to select a dramatically smaller number of subgraphs. Furthermore, we demonstrate its advantages over random selection baselines and previous methods on a wide range of tasks.

**Contributions.** This paper provides the following contributions: (1) A new framework for learning subgraph selection policies; (2) A theoretical analysis motivating the policy learning task as well as our architecture; and (3) An experimental evaluation of the new approach demonstrating its advantages in terms of predictive performance and reduced runtime.

## 2 RELATED WORK

**Expressive power limitations of GNNs.** Multiple works have been proposed to overcome the expressive power limitations of MPNNs, which are constrained by the WL isomorphism test (Xu et al., 2019; Morris et al., 2019). Other than subgraph-based methods, which are the focus of this work, these approaches include: (1) Architectures aligned to the $k$-WL hierarchy (Morris et al., 2019; 2020b; Maron et al., 2019b;a); (2) Augmentations of the node features with node identifiers (Abboud et al., 2020; Sato, 2020; Dasoulas et al., 2021; Murphy et al., 2019); (3) Models leveraging additional information, such as homomorphism and subgraph counts (Barceló et al., 2021; Bouritsas et al., 2022), graph polynomials (Puny et al., 2023), and simplicial and cellular complexes (Bodnar et al., 2021b;a). We refer the reader to the survey by Morris et al. (2021) for a review of different methods.

**Subgraph GNNs.** Despite the differences in the specific layer operations, the idea of constructing bags of subgraphs to be processed through message-passing layers has been recently proposed by several concurrent works. Cotta et al. (2021) proposed to first obtain subgraphs by deleting nodes in the original graph, and then apply an MPNN to each subgraph separately, whose representations are finally aggregated through a set function. Zhang & Li (2021) considered subgraphs obtained by constructing the local ego-network around each node in the graph, an approach also followed by Zhao et al. (2022), which further incorporates feature aggregation modules that aggregate node representations across different subgraphs. Bevilacqua et al. (2022) developed two classes of Subgraph GNNs, which differ in the symmetry group they are equivariant to: (i) DS-GNN, which processes each subgraph independently, and (ii) DSS-GNN, which includes feature aggregation modules of the same node in different subgraphs. Huang et al. (2023); Yan et al. (2023); Papp & Wattenhofer (2022) further studied the expressive power of Subgraph GNNs, while Zhou et al. (2023) proposed a framework that encompasses several existing methods. Recently, Frasca et al. (2022); Qian et al. (2022) provided a theoretical analysis of node-based Subgraph GNNs, demonstrating that they are upper-bounded by 3-WL, while Zhang et al. (2023) presented a complete hierarchy of existing architectures with growing expressive power. Finally, other works such as Rong et al. (2019); Vignac et al. (2020); Papp et al. (2021); You et al. (2021) can be interpreted as Subgraph GNNs.

Most related to our work is the recent paper of Qian et al. (2022), which, to the best of our knowledge, presented the first framework that learns a subgraph selection policy rather than employing a full bag or a random selection policy. Our work differs from theirs in four main ways: (i) Bag of subgraphs construction: Our method generates the bag sequentially, compared to their one-shot generation; (ii) Expressive selection GNN architecture: We parameterize the selection network using a Subgraph GNN, that is more expressive than the MPNN used by Qian et al. (2022); (iii) Differentiable sampling mechanism: We use the well-known Gumbel-Softmax trick to enable gradient backpropagation through the discrete sampling process, while Qian et al. (2022) use I-MLE (Niepert et al., 2021); and (iv) Motivation: our work is primarily motivated by scenarios where a subset of subgraphs is sufficient for maximal expressive power, while Qian et al. (2022) aim to make higher-order generation policies (where each subgraph is generated from tuples of nodes) practical. We prove that our choices lead to a framework that can learn subgraph distributions that cannot be expressed with Qian et al. (2022), and show that our framework performs better than Qian et al. (2022) on real-world datasets.

## 3 PROBLEM FORMULATION

Let $G = (A, X)$, be a node-attributed, undirected, simple graph with $n$ nodes, where $A \in \mathbb{R}^{n \times n}$ represents the adjacency matrix of $G$ and $X \in \mathbb{R}^{n \times c}$ is the node feature matrix. Given a graph $G$, a Subgraph GNN first transforms $G$ into a bag (multiset) of subgraphs, $\mathcal{B}_G = \{\!\{ S_1, S_2, \ldots, S_m \}\!\}$, obtained from a predefined generation policy, where $S_i = (A_i, X_i)$ denotes subgraph $i$. Then, it processes $\mathcal{B}_G$ through a stacking of (node- and subgraph-) permutation equivariant layers, followed by a pooling function to obtain graph or node representations to be used for the final predictions.

In this paper, we focus on the node marking (NM) subgraph generation policy (Papp & Wattenhofer, 2022), where each subgraph is obtained by marking a single node in the graph, without altering the connectivity of the original graph[1]. More specifically, for each subgraph $S_i = (A_i, X_i)$, there exists a node $v_i$ such that $A_i = A$ and $X_i = X \oplus \chi_{v_i}$, where $\oplus$ denotes channel-wise concatenation and

---

[1]Although technically, NM does not generate subgraphs in the traditional sense, the marked versions of the original graph are commonly regarded as subgraphs (Papp & Wattenhofer, 2022).

$\chi_{v_i}$ is a one-hot indicator vector for the node $v_i$. We refer to the node $v_i$ as the root of $S_i$, which we will equivalently denote as $S_{v_i}$. Notably, since each subgraph is obtained by marking exactly one node, there is a bijection between nodes and subgraphs, making the total number of subgraphs under the NM policy equal to the number of nodes $m = n$. Finally, we remark that we chose NM as the generation policy as it generalizes several other policies (Zhang et al., 2023). Any other node-based policy, preserving this bijection between nodes and subgraphs, can be equivalently considered.

**Objective.** Motivated by their recent success, we would like to employ a Subgraph GNN to solve a given graph learning task (e.g., graph classification). Aiming to mitigate the computational overhead of using the full bag $\mathcal{B}_G$, we wish to propose a method for learning a subgraph selection policy $\pi(G)$ that, given the input graph $G$, returns a subset of $\mathcal{B}_G$ to be used as an input to the Subgraph GNN in order to solve the downstream task at hand.

We denote by $\mathcal{B}_G^T \subset \mathcal{B}_G$ the output of $\pi(G)$, consisting of the original graph and $T$ *selected* subgraphs. That is, $\mathcal{B}_G^T = \{\!\!\{ G, S_{v_1}, S_{v_2}, \ldots, S_{v_T} \}\!\!\}$, where node $v_j$ in $G$ is the root node of $S_{v_j}$.

## 4 INSIGHTS FOR THE SUBGRAPH SELECTION LEARNING PROBLEM

In this section, we motivate the problem of learning to select subgraphs in the context of Subgraph GNNs. Specifically, we seek to address two research questions: (i) *Are there subgraph selection policies that return small yet effective bags of subgraphs?*, and (ii) *How do these policies compare to strategies that uniformly sample a random subset of all possible subgraphs in the bag?*. In the following, we present our main results, while details and proofs are provided in Appendix C.

**Powerful policies containing only a single subgraph.** To address the first question, we consider Circulant Skip Links (CSL) graphs, an instantiation of which was shown in Figure 1 (CSL(13, 5) and CSL(13, 3)).

**Definition 1** (CSL graph (Murphy et al., 2019)). *Let $n$, $k$, be such that $n$ is co-prime with $k$ and $k < n - 1$. A CSL graph, CSL(n, k) is an undirected graph with $n$ nodes labeled as $\{0, \ldots, n-1\}$, whose edges form a cycle and have skip links. That is, the edge set $E$ is defined by a cycle formed by $(j, j + 1) \in E$ for $j \in \{0, \ldots, n-2\}$, and $(n-1, 0) \in E$, along with skip links defined recursively by the sequence $(s_i, s_{i+1}) \in E$, with $s_1 = 0$ and $s_{i+1} = (s_i + k) \bmod n$.*

While CSL graphs are WL-indistinguishable (Murphy et al., 2019), they become distinguishable when considering the full bag of node-marked subgraphs (Cotta et al., 2021). Furthermore, since all subgraphs in the bag are isomorphic, it is easy to see that distinguishability is obtained even when considering a single subgraph per graph. This observation motivates the usage of small bags, as there exist cases where the expressive power of the full bag can be obtained with a very small subset of subgraphs.

The extreme case of CSL graphs, however, is too simplistic: randomly selecting one subgraph in each graph is sufficient for disambiguation. This leads to our second question, and, in the following, we build upon CSL graphs to define a family of graphs where random policies are not effective. As we shall see next, in these families only specific small subgraph selections lead to complete identification of the isomorphism type of each graph.

**Families of graphs that require specific selections of $\ell$ subgraphs.** We obtain our family of graphs starting from CSL graphs, which we use as building blocks to define an $(n, \ell)$-CSL graph.

**Definition 2** ($(n, \ell)$-CSL graph). *A $(n, \ell)$-CSL graph, denoted as $(n, \ell)$-CSL(n, $(k_1, \ldots, k_\ell)$), is a graph with $\ell \cdot n$ nodes obtained from $\ell$ disconnected non-isomorphic CSL (sub-)graphs with $n$ nodes, CSL(n, $k_i$), $i \in \{1, \ldots, \ell\}$. Note that the maximal value of $\ell$ depends on $n$.*

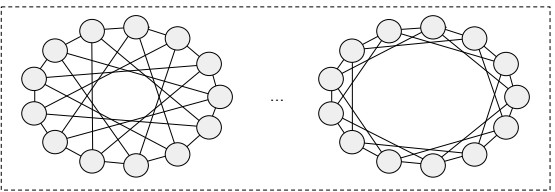

Figure 2: An $(n, \ell)$-CSL graph is obtained from $\ell$ disconnected, non-isomorphic CSL graphs with $n$ nodes, where $n = 13$ in this case.

Our interest in these graphs lies in the fact that the family of non-isomorphic $(n, \ell)$-CSL graphs contains WL-indistinguishable graphs, as we show in the following.

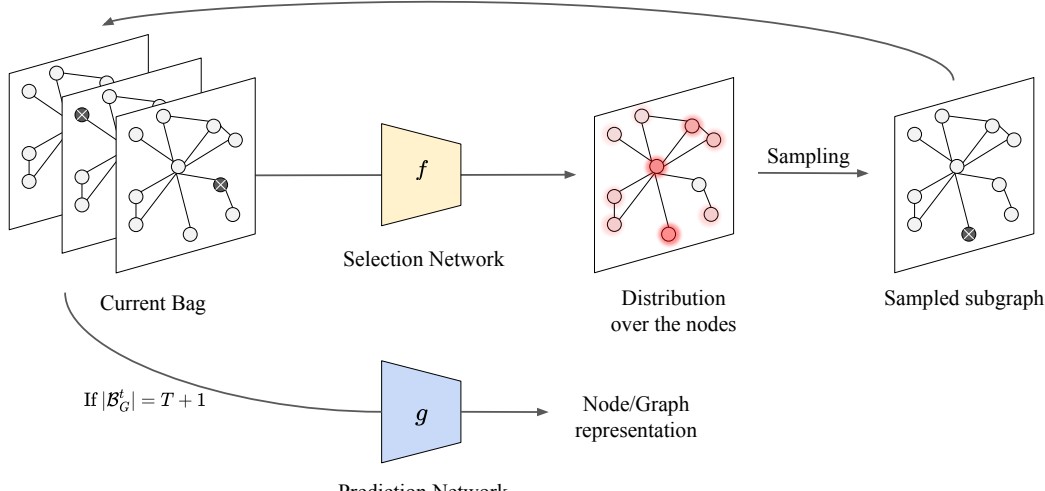

Figure 3: An overview of POLICY-LEARN. POLICY-LEARN consists of two Subgraph GNNs: a selection network and a prediction network. The selection network generates the bag of subgraphs by iteratively parameterizing a probability distribution over the nodes of the original graph. When the bag size reaches its maximal size $T + 1$ (the original graph plus $T$ selections), it is passed to the prediction network for the downstream task.

**Theorem 1** $((n, \ell)$-CSL graphs are WL-indistinguishable)**.** *Let $\mathcal{G}_{n,\ell}$ be the family of* non-isomorphic $(n, \ell)$-*CSL graphs (Definition 2). All graphs in $\mathcal{G}_{n,\ell}$ are WL-indistinguishable.*

While these graphs are WL-indistinguishable, we can distinguish all pairs when utilizing the full bag of node-marked subgraphs. More importantly, the full bag allows us to fully identify the isomorphism type of all graphs within the family, which implies the distinguishability of pairs of graphs[2]. As we show next, it is however unnecessary to consider the entire bag of all the $\ell \cdot n$ subgraphs. Instead, there exists a subgraph selection policy that returns only $\ell$ subgraphs, a significantly smaller number than the total in the full bag, which are provably sufficient for identifying the isomorphism type of all graphs within the family. This identification is attained when the root nodes of all $\ell$ subgraphs belong to different CSL (sub-)graphs. Furthermore, we prove that this condition is not only sufficient but also necessary. In other words, the isomorphism-type identification of all graphs within the family only occurs when there is at least one node-marked subgraph obtained from each CSL (sub-)graph.

**Proposition 2** (There exists an efficient $\pi$ that fully identifies $(n, \ell)$-CSL graphs)**.** *Let $\mathcal{G}_{n,\ell}$ be the family of non-isomorphic $(n, \ell)$-CSL graphs (Definition 2). A node-marking based subgraph selection policy $\pi$ can identify the isomorphism type of any graph $G$ in $\mathcal{G}_{n,\ell}$ if and only if its bag has a marking in each CSL (sub-)graph.*

The above results demonstrate the existence of efficient and more sophisticated subgraph selection policies that enable the identification of isomorphism types. Remarkably, these can return bags as small as containing $\ell$ subgraphs, as long as each of them is obtained by marking a node in a different CSL (sub-)graph. The insight derived from the necessary condition is equally profound: a random selection approach, which uniformly samples $\ell$ subgraphs per graph, does not ensure identification of the isomorphism type, because the subgraphs might not be drawn from distinct CSL (sub-)graphs. This finding highlights the sub-optimality of random selection strategies, as we formalize below.

**Proposition 3** (A random policy cannot efficiently identify $(n, \ell)$-CSL graphs)**.** *Let $G$ be any graph in the family of non-isomorphic $(n, \ell)$-CSL graphs, namely $\mathcal{G}_{n,\ell}$ (Definition 2). The probability that a random policy, which uniformly samples $\ell$ subgraphs from the full bag, can successfully identify the isomorphism type of $G$ is $\ell!/\ell^\ell$, which tends to 0 as $\ell$ increases.*

The above result has important implications for Subgraph GNNs when coupled with random policies. It indicates that Subgraph GNNs combined with a random subgraph selection policy, are unlikely to

---

[2]Note that we consider the family of non-isomorphic $(n, \ell)$-CSL graphs, thus excluding cases where the choices of $k_1, \ldots, k_\ell$ lead to isomorphic connected components.

---

**Algorithm 1** POLICY-LEARN: Feedforward with learnable subgraph selection policy

---

1: Initialize bag with original graph: $\mathcal{B}_G^0 := \{\!\{G\}\!\}$
2: **for** $t = 0, \ldots, T - 1$ **do**
3:     Feed current bag $\mathcal{B}_G^t$ to selection network $f$, obtain a distribution over the nodes of $G$, $p_{t+1} = f(\mathcal{B}_G^t)$
4:     Sample node $v_{t+1} \sim p_{t+1}$
5:     Add subgraph to bag $\mathcal{B}_G^{t+1} = \mathcal{B}_G^t \cup \{\!\{S_{v_{t+1}}\}\!\}$
6: **end for**
7: Feed final bag $\mathcal{B}_G^T$ to the downstream network $g$ and output result $g(\mathcal{B}_G^T)$

---

identify the isomorphism type of $G$, assuming the number of sampled subgraphs is $\ell$. In Lemma 6 in Appendix C, we further show that the expected number of subgraphs that a random policy must uniformly draw before identification is $T = \Theta(\ell \ln \ell)$. Notably, this number is significantly larger than the minimum number $\ell$ of subgraphs required by $\pi$. For instance, when $\ell = 100$, a random policy would have to sample around $460$ subgraphs per graph. This is more than four times the number of subgraphs returned by $\pi$, making it clear that there exists a substantially more effective approach than random sampling to the problem.

## 5 METHOD

The following section describes the proposed framework: POLICY-LEARN. We start with an overview of the method, before discussing in detail the specific components, namely the selection and the prediction networks, as well as the learning procedure used to differentiate through the discrete sampling process.

### 5.1 OVERVIEW

Our method is illustrated in Figure 3 and described in Algorithm 1. It is composed of two main Subgraph GNNs: a subgraph selection network, denoted as $f$, that generates the subgraph selection policy $\pi$, and a downstream prediction network, denoted as $g$, that solves the learning task of interest. More concretely, given an input graph $G$, we initialize the bag of subgraphs with the original graph, $\mathcal{B}_G^0 := \{\!\{G\}\!\}$. Then, at each step $t + 1$, the selection network $f$ takes the current bag of subgraphs $\mathcal{B}_G^t$ as input and outputs an un-normalized distribution $p_{t+1} := f(\mathcal{B}_G^t) \in \mathbb{R}^n$, defined over the nodes of the original graph $G$. Our algorithm samples from $p_{t+1}$ to select a node $v_{t+1}$ that will serve as the root of the next subgraph $S_{v_{t+1}}$. This new subgraph is added to the bag, i.e. we define $\mathcal{B}_G^{t+1} = \mathcal{B}_G^t \cup \{\!\{S_{v_{t+1}}\}\!\}$. After $T$ subgraph selections, where $T$ is a hyperparameter representing the number of subgraphs in the bag ($T + 1$ considering the original graph), the final bag $\pi(G) := \mathcal{B}_G^T$ is passed to the prediction network $g$, a Subgraph GNN designed to solve the downstream task at hand. Both $f$ and $g$ are trained jointly in an end-to-end manner, and we use the Gumbel-Softmax trick (Jang et al., 2016; Maddison et al., 2017) to backpropagate through the discrete sampling process. In what follows, we elaborate on the exact implementation and details of each component of our approach.

### 5.2 SUBGRAPH SELECTION NETWORK

The goal of $f$ is to output a distribution over the nodes of the original graph, determining which node should serve as the root of the next subgraph that will be added to the bag. As this network takes a bag of subgraphs as its input, it is natural to implement $f$ as a Subgraph GNN. Specifically, we employ the most efficient Subgraph GNN variant, namely DS-GNN (Cotta et al., 2021; Bevilacqua et al., 2022), that performs message-passing operations on each subgraph independently using an MPNN, referred to as the subgraph encoder. Subsequently, it applies a pooling operation across the subgraphs in the bag, to obtain a single representation for each node in the input graph $G$.

Since we have both the original node features and the markings, the subgraph encoder in our DS-GNN architecture is implemented using the model proposed by Dwivedi et al. (2021), which decouples the two representations to improve learning. This encoder consists of two message-passing functions: $f_p$, responsible for updating the markings, and $f_h$, responsible for updating the original features. Formally, at the $t$-th iteration, we are provided with a bag of subgraphs $\mathcal{B}_G^t$ of size $t + 1$. For each

subgraph $s = 0, \ldots, t$, we compute the following for $l = 1, \ldots, L_1$ where $L_1$ represents the total number of layers in the selection network:

$$p_s^{(l)} = f_p^{(l)}(p_s^{(l-1)}, A_s), \ p_s^{(l)} \in \mathbb{R}^{n \times c}, \tag{1a}$$

$$h_s^{(l)} = f_h^{(l)}(h_s^{(l-1)} \oplus p_s^{(l-1)}, A_s), \ h_s^{(l)} \in \mathbb{R}^{n \times c}, \tag{1b}$$

with $f_p^{(l)}$ and $f_h^{(l)}$ the $l$-th message passing layers in $f_p$ and $f_h$, respectively. At $l = 0$, we initialize the marking as $p_s^{(0)} = \chi_{v_s} \in \mathbb{R}^n$ and the input node features as $h_s^{(0)} = X \in \mathbb{R}^{n \times c_{in}}$.

After the $L_1$-th layer, we combine the marking and node features to obtain the updated node features, as follows:

$$h_s = p_s^{(L_1)} + h_s^{(L_1)}, \ h_s \in \mathbb{R}^{n \times c}. \tag{2}$$

Note that in this way we obtain node features for every subgraph in the bag. In order to unify them into a single output per node, we pool the node features along the subgraph dimension. We then apply an MLP that reduces the channel dimension to one, to obtain the following un-normalized node probabilities from which we sample the next root node:

$$p_{t+1} = h_{\mathcal{B}_G^t} = \text{MLP}(\text{pool}(\{\!\{h_0, \ldots, h_t\}\!\})), \ p_{t+1} \in \mathbb{R}^n. \tag{3}$$

**Backpropagating through the sampling process.** A naive sampling strategy is not differentiable and therefore will prevent us from backpropagating gradients to update the weights of the selection network $f$. Therefore, we use the straight-through Gumbel-Softmax estimator (Jang et al., 2016; Maddison et al., 2017) to sample from $p_{t+1}$, and obtain the one-hot encoding marking of the next subgraph to be added to the current bag of subgraphs:

$$\chi_{v_{t+1}} = \text{GumbelSoftmax}(p_{t+1}), \ \chi_{v_{t+1}} \in \{0,1\}^n. \tag{4}$$

We denote the subgraph associated with the node marking $\chi_{v_{t+1}}$ by $S_{v_{t+1}}$. Therefore, the bag of subgraphs at the $(t+1)$-th iteration reads $\mathcal{B}_G^{t+1} = \mathcal{B}_G^t \cup \{\!\{S_{v_{t+1}}\}\!\}$.

## 5.3 Downstream Prediction Network

Similarly to the selection network $f$, our prediction network $g$ is parameterized as a Subgraph GNN. It takes as input the final bag of subgraphs $\mathcal{B}_G^T$ that was sampled using $f$, and produces graph- or node-level predictions depending on the task of interest. Like the selection network, we employ a DS-GNN architecture and implement the subgraph encoder using two message-passing functions, $g_p$ and $g_h$, with layers $l = 1, \ldots, L_2$ and updates identical to Equations (1a) and (1b). Note that, while $f$ and $g$ are architecturally similar, they are two different networks with their own set of learnable weights. In the first layer, i.e., $l = 0$, we set the initial marking as $\bar{p}_s^{(0)} = \chi_{v_s} \in \mathbb{R}^n$ and input node features as $\bar{h}_s^{(0)} = X \in \mathbb{R}^{n \times c_{in}}$, where we use the bar, i.e., $\bar{p}_s^{(l)}, \bar{h}_s^{(l)}$, to denote the representations in the downstream prediction model and distinguish them from the ones in the selection model.

After the $L_2$-th layer, we combine the marking and node features to obtain the updated node features $\bar{h}_s = \bar{p}_s^{(L_2)} + \bar{h}_s^{(L_2)}$ and pool the node features along the subgraph and potentially the node dimension to obtain a final node or graph representation. This representation is then further processed with an MLP to obtain the final prediction.

## 6 Theoretical Analysis

In this section, we study the theoretical capabilities of our framework, focusing on its ability to provably identify the isomorphism type of the $(n, \ell)$-CSL graphs introduced in Section 4. Due to space constraints, all proofs are relegated to Appendix C. We start by showing that, differently than random selection approaches, POLICY-LEARN can identify the isomorphism type of any $(n, \ell)$-CSL graph (Definition 2) when learning to select only $\ell$ subgraphs. Remarkably, this number is significantly smaller than the total $\ell \cdot n$ subgraphs in the full bag, as well as the expected number of subgraphs $\Theta(\ell \ln \ell)$ required by a random policy.

**Theorem 4** (POLICY-LEARN can identify $\mathcal{G}_{n,\ell}$). *Let $G$ be any graph in the family of non-isomorphic $(n, \ell)$-CSL graphs, namely $\mathcal{G}_{n,\ell}$ (Definition 2). There exist weights of $f$ and $g$ such that POLICY-LEARN with $T = \ell$ can identify the isomorphism type of $G$ within the family.*

The idea of the proof lies in the ability of the selection network within POLICY-LEARN to implement a policy $\pi$ that returns $\ell$ subgraphs per graph where: (i) each subgraph is obtained by marking exactly one node and (ii) each marked node belongs to a different CSL (sub-)graph. Given that this policy serves as a sufficient condition for the identification of the isomorphism types (Proposition 2), our proposed framework successfully inherits this ability. More specifically, the proof (in Appendix C) proceeds by showing that the selection network can always distinguish all the nodes belonging to already-marked CSL (sub-)graphs from the rest, and therefore can assign them a zero probability of being sampled, while maintaining a uniform probability on all the remaining nodes.

The previous result further sets our method apart from the OSAN method proposed in Qian et al. (2022). Indeed, as its selection network is parameterized as an MPNN, and given that $\mathcal{G}_{n,\ell}$ contains WL-indistinguishable graphs composed by WL-indistinguishable CSL (sub-)graphs, then OSAN cannot differentiate between all the nodes in the graph. Therefore, since all the subgraphs are selected simultaneously rather than sequentially, the resulting sampling process is the same as uniform sampling. Consequently, it cannot ensure that marked nodes belong to different CSL (sub-)graphs. Since this condition is not only sufficient but also necessary for the identification of the isomorphism type (Proposition 2), OSAN effectively behaves like a random policy for indistinguishable graphs.

**Theorem 5** (Qian et al. (2022) cannot efficiently identify $\mathcal{G}_{n,\ell}$). *Let $G$ be any graph in the family of non-isomorphic $(n, \ell)$-CSL graphs, namely $\mathcal{G}_{n,\ell}$ (Definition 2). Then, the probability that OSAN (Qian et al., 2022) with $T = \ell$ subgraphs can successfully identify the isomorphism type of $G$ is $\ell!/\ell^\ell$, which tends to 0 as $\ell$ increases.*

Up until this point, our focus has been on demonstrating that POLICY-LEARN has the capability to implement the policy $\pi$ for identification within $\mathcal{G}_{n,\ell}$. However, it is important to note that $\pi$ is not the sole policy that POLICY-LEARN can implement. Indeed, POLICY-LEARN can also learn subgraph selection policies that depend on the local graph structure, such as the policy returning all subgraphs with root nodes having degree greater than a specified threshold (see Proposition 7 in Appendix C).

## 7 EXPERIMENTS

In this section, we empirically investigate the performance of POLICY-LEARN. In particular, we seek to answer the following questions: (1) Can our approach consistently outperform MPNNs, as demonstrated by full-bag Subgraph GNNs? (2) Does POLICY-LEARN achieve better performance than a random baseline that uniformly samples a subset of subgraphs at each training epoch? (3) How does our method compare to the existing learnable selection strategy OSAN (Qian et al., 2022)? (4) Is our approach competitive to full-bag Subgraph GNNs on real-world datasets?

In the following, we present our main results and refer to Appendix E for details.[3] For each task, we include two baselines: RANDOM, which represents the random subgraph selection, and FULL, which corresponds to the full-bag approach. These baselines utilize the same downstream prediction network (appropriately tuned) as POLICY-LEARN, with RANDOM selecting random subsets of subgraphs equal in size to those used by POLICY-LEARN (denoted by $T$), and FULL using the entire bag.

**ZINC.** We experimented with the ZINC-12K molecular dataset (Sterling & Irwin, 2015; Gómez-Bombarelli et al., 2018; Dwivedi et al., 2020), where as prescribed we maintain a 500k parameter budget. As can be seen from Table 1, POLICY-LEARN significantly improves over OSAN, and surpasses the random baseline. Notably, OSAN performs worse than our random baseline due to differences in the implementation of the prediction network. As expected, the gap between POLICY-LEARN and RANDOM is larger when the number of subgraphs $T$ is smaller, where selecting the most informative subgraphs is more crucial.

Table 1: Test results on the ZINC molecular dataset under the 500k parameter budget.

| Method | ZINC (MAE ↓) |
| --- | --- |
| GCN (Kipf & Welling, 2017) | 0.321±0.009 |
| GIN (Xu et al., 2019) | 0.163±0.004 |
| PNA (Corso et al., 2020) | 0.133±0.011 |
| GSN (Bouritsas et al., 2022) | 0.101±0.010 |
| CIN (Bodnar et al., 2021a) | 0.079±0.006 |
| OSAN (Qian et al., 2022) $T = 2$ | 0.177±0.016 |
| FULL | 0.087±0.003 |
| RANDOM $T = 2$ | 0.136±0.005 |
| POLICY-LEARN $T = 2$ | 0.120±0.003 |
| RANDOM $T = 5$ | 0.113±0.006 |
| POLICY-LEARN $T = 5$ | 0.109±0.005 |

---

[3] Our code is available at `https://github.com/beabevi/policy-learn`

Table 3: Test results on the OGB datasets. POLICY-LEARN outperforms both the learnable selection approach OSAN and the random selection by a large margin, while approaching or surpassing the corresponding full-bag architecture. N/A indicates the result was not reported.

| Method ↓ / Dataset → | MOLESOL RMSE ↓ | MOLTOX21 ROC-AUC ↑ | MOLBACE ROC-AUC ↑ | MOLHIV ROC-AUC ↑ |
|---|---|---|---|---|
| GCN (Kipf & Welling, 2017) | 1.114±0.036 | 75.29±0.69 | 79.15±1.44 | 76.06±0.97 |
| GIN (Xu et al., 2019) | 1.173±0.057 | 74.91±0.51 | 72.97±4.00 | 75.58±1.40 |
| OSAN (Qian et al., 2022) $T = 10$ | 0.984±0.086 | N/A | 72.30±6.60 | N/A |
| FULL | 0.847±0.015 | 76.25±1.12 | 78.41±1.94 | 76.54±1.37 |
| RANDOM $T = 2$ | 0.951±0.039 | 76.65±0.89 | 75.36±4.28 | 77.55±1.24 |
| POLICY-LEARN $T = 2$ | 0.877±0.029 | 77.47±0.82 | 78.40±2.85 | 79.13±0.60 |
| RANDOM $T = 5$ | 0.900±0.032 | 76.62±0.63 | 78.14±2.36 | 77.30±2.56 |
| POLICY-LEARN $T = 5$ | 0.883±0.032 | 77.36±0.60 | 78.39±2.28 | 78.49±1.01 |

**Reddit.** To showcase the capabilities of POLICY-LEARN on large graphs, we experimented with the REDDIT-BINARY dataset (Morris et al., 2020a) where full-bag Subgraph GNNs cannot be otherwise applied due to the high average number of nodes, and consequently subgraphs, per graph (429.63). POLICY-LEARN exhibits the best performance among all efficient baselines (Table 2), while maintaining competitive inference runtimes (Table 7 in Appendix E), opening up the applicability of Subgraph GNNs to unexplored avenues. Table 6 in Appendix E further compares to random baselines coupled with different subgraph-based downstream models.

Table 2: Results on the REDDIT-BINARY dataset demonstrate the efficacy on large graphs.

| Method | RDT-B (ACC ↑) |
|---|---|
| GIN (Xu et al., 2019) | 92.4±2.5 |
| SIN (Bodnar et al., 2021b) | 92.2±1.0 |
| CIN (Bodnar et al., 2021a) | 92.4±2.1 |
| FULL | OOM |
| RANDOM $T = 20$ | 92.6±1.5 |
| RANDOM $T = 2$ | 92.4±1.0 |
| POLICY-LEARN $T = 2$ | 93.0±0.9 |

**OGB.** We tested our framework on several datasets from the OGB benchmark collection (Hu et al., 2020). Table 3 shows the performances of our method when compared to MPNNs and efficient subgraph selection policies. For OSAN, we report the best result across all node-based policies, which is obtained with a larger number of subgraphs than what we consider ($T = 10$ compared to our $T \in \{2, 5\}$). Notably, POLICY-LEARN improves over MPNNs while retaining similar asymptotic complexity, and consistently outperforms OSAN and RANDOM. The performance gap between POLICY-LEARN and RANDOM is more significant with $T = 2$, particularly on MOLESOL and MOLBACE. Remarkably, POLICY-LEARN approaches the full-bag baseline FULL on MOLESOL and MOLBACE, and even outperforms it on MOLTOX21 and MOLHIV, achieving the highest score. These results can be attributed to the generalization challenges posed by the dataset splits, which are further confirmed by the observation that increasing the number of subgraphs does not improve results. Table 4 in Appendix E additionally reports the results of existing full-bag Subgraph GNNs.

## 8 CONCLUSIONS

In this paper we introduced a novel framework, POLICY-LEARN, for learning to select a small subset of the large set of possible subgraphs in order to reduce the computational overhead of Subgraph GNNs. We addressed the pivotal question of whether a small, carefully selected, subset of subgraphs can be used to identify the isomorphism type of graphs that would otherwise be WL-indistinguishable. We proved that, unlike random selections and previous work, our method can provably learn to select these critical subgraphs. Empirically, we demonstrated that POLICY-LEARN consistently outperforms random selection strategies and approaches the performance of full-bag methods. Furthermore, it significantly surpasses prior work addressing the same problem, across all the datasets we considered. This underscores the effectiveness of our framework in focusing on the necessary subgraphs, mitigating computational complexity, and achieving competitive results. We believe our approach opens the door to broader applications of Subgraph GNNs.

ACKNOWLEDGMENTS

BR acknowledges support from the National Science Foundation (NSF) awards CCF-1918483, CA-REER IIS-1943364 and CNS-2212160, Amazon Research Award, AnalytiXIN, the Wabash Heartland Innovation Network (WHIN), Ford, NVidia, CISCO, and Amazon. Computing infrastructure was supported in part by CNS-1925001 (CloudBank). HM is the Robert J. Shillman Fellow, and is supported by the Israel Science Foundation through a personal grant (ISF 264/23) and an equipment grant (ISF 532/23). Any opinions, findings and conclusions or recommendations are those of the authors and do not necessarily reflect the views of the sponsors.

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

## A  ADDITIONAL RELATED WORK

Our approach is related not only to Subgraph GNNs and expressive GNN architecture, as discussed in the main paper, but also to other sampling strategies in the graph domain. In the context of active learning, Hu et al. (2020) propose a reinforcement learning approach that learns which subset of nodes to label in order to reduce the annotation cost of training GNNs. For controlling the dynamics of a system using localized interventions, Meirom et al. (2021) employ a reinforcement learning agent that selects a subset of nodes at each step and attempts to change their state, with an objective that depends on the number of nodes in each state. More similar to our sampling strategy is the work by Martinkus et al. (2023) which proposes a GNN architecture augmented with *neural agents*, that traverse the graph and then collectively classify it. Notably, at each step, an agent chooses a neighboring node to transition to using the Gumbel-Softmax trick.

## B  ILLUSTRATION OF WL ON THE BAG OF SUBGRAPHS

In this section we explain in detail Figure 1. For each original graph, we consider the application of the WL test to the subgraph obtained by marking the crossed node. More precisely, at time step 0, each non-marked node in the subgraph is assigned the same initial constant color, which differs from the initial color of the marked node. Without loss of generality, denote these colors as color 1 and color 0, respectively. Then, the WL test proceeds by refining the color of each node by aggregating the colors of its neighbors (including itself). More precisely, the new color of a node is obtained through a hash function taking as input the multiset of colors of the neighbors and of the node itself. It is easy to see that, at time step 1, the marked node has a unique color, which we call color 2, its neighbors all have the same colors, color 3, and all the remaining nodes have the same color (different from the marked node and its neighbors), color 4. The refinement is repeated until we reach a stable coloring. At this point, we simply collect the number of nodes having the same color in a histogram. For the upper graph in Figure 1, the histogram indicates that in the stable coloring, there are 4 blue nodes, 4 green nodes, 4 yellow nodes, and 1 orange node. Importantly, since the WL test represents a necessary condition for graph isomorphism, different histograms imply that the two graphs are not isomorphic, and thus the WL test distinguishes them. Figure 1 shows that marking only one node is sufficient for distinguishing the two graphs, which are instead indistinguishable when no node is marked, as the WL test returns the same histogram for them (13 orange nodes).

## C  PROOFS

This appendix includes the proofs for the theoretical results presented in Sections 4 and 6. We start by formally proving that the family $\mathcal{G}_{n,\ell}$ of non-isomorphic $(n, \ell)$-CSL graphs, defined in Definition 2, contains WL-indistinguishable graphs.

**Theorem 1** ($(n, \ell)$-CSL graphs are WL-indistinguishable). *Let $\mathcal{G}_{n,\ell}$ be the family of* non-isomorphic $(n, \ell)$-*CSL graphs (Definition 2). All graphs in $\mathcal{G}_{n,\ell}$ are WL-indistinguishable.*

*Proof.* Let $G_1$, $G_2$ be any two $(n, \ell)$-CSL graphs in $\mathcal{G}_{n,\ell}$. We will prove that $G_1$, $G_2$ are WL-indistinguishable by simulating one WL round starting from a constant color initialization.

(Iter. 1)  The colors after the first iteration represent the node degrees. Since each node has degree 4 (because it belongs to one CSL (sub-)graph), then all nodes will have the same color.

Since the colors after the first iterations are a finer refinement of the initialization colors, the algorithm stops and the graphs are not separated. □

Next, we show that there exists an efficient policy $\pi$, that can return as few subgraphs as $\ell$, providing strong guarantees. Indeed, $\pi$ represents a necessary and sufficient condition for identifying the isomorphism type of all graphs in $\mathcal{G}_{n,\ell}$, as we prove next.

**Proposition 2** (There exists an efficient $\pi$ that fully identifies $(n, \ell)$-CSL graphs). *Let $\mathcal{G}_{n,\ell}$ be the family of non-isomorphic $(n, \ell)$-CSL graphs (Definition 2). A node-marking based subgraph selection*

*policy $\pi$ can identify the isomorphism type of any graph $G$ in $\mathcal{G}_{n,\ell}$ if and only if its bag has a marking in each CSL (sub-)graph.*

*Proof.* We will prove the two cases separately.

**Sufficiency.** Consider any graph $G \in \mathcal{G}_{n,\ell}$, and let $S_i$ be a subgraph returned by $\pi$ obtained by marking any node in the $i$-th CSL (sub-)graph of graph $G$. Note that $S_i$ is sufficient to identify the isomorphism type of the $i$-th CSL (sub-)graph, because marking any node is sufficient for disambiguation of any possible pair of CSL graphs (Cotta et al., 2021, Theorem 2). Consider the set $H$ of $\ell$ histograms, each obtained by marking a different CSL (sub-)graph. Note that if $\pi$ returns more than $\ell$ subgraphs, then identical histograms indicate that the corresponding subgraphs are obtained from the same CSL (sub-)graph, because $G$ is composed by non-isomorphic CSL (sub-)graphs. Thus, identical histograms can be simply discarded when constructing $H$. Then, the set $H$ is sufficient to identify the isomorphism type of $G$, as each element of $H$ identifies one of its CSL (sub-)graphs.

**Necessity.** The proof proceeds by contrapositive. Consider a policy $\pi'$ and assume that there exists at least one CSL (sub-)graph such that none of the marked nodes belongs to it. Without loss of generality, assume that only one CSL (sub-)graphs is not covered, meaning no marked nodes belong to it. Then, $\pi'$ cannot identify the isomorphism type of $G$, because it is unaware of the isomorphism type of the CSL (sub-)graph that is not covered. More precisely, consider a graph $G' \in \mathcal{G}_{n,\ell}$ such that: (1) $G'$ is non-isomorphic to $G$ and (2) $G'$ differ from $G$ only in one CSL (sub-)graph, which is the one not covered by $\pi'$. Then, $\pi'$ cannot identify whether the isomorphism type of its input is the one of $G$ or the one of $G'$. $\square$

The above theorem implies that a random selection baselines might fail to select the subgraphs required for the identification, as it might fail to draw at least one subgraph from each CSL (sub-)graph. In the following we prove that the implications are even more severe as $\ell$ grows.

**Proposition 3** (A random policy cannot efficiently identify $(n, \ell)$-CSL graphs)**.** *Let $G$ be any graph in the family of non-isomorphic $(n, \ell)$-CSL graphs, namely $\mathcal{G}_{n,\ell}$ (Definition 2). The probability that a random policy, which uniformly samples $\ell$ subgraphs from the full bag, can successfully identify the isomorphism type of $G$ is $\ell!/\ell^\ell$, which tends to 0 as $\ell$ increases.*

*Proof.* From Proposition 2, each of the $\ell$ subgraphs must be generated by marking a node belonging to a different CSL (sub-)graph for the identification of the isomorphism type (necessary condition). Recall that the total number of CSL (sub-)graphs is $\ell$, and each CSL (sub-)graph has exactly $n$ nodes. Thus, the probability of choosing $\ell$ nodes from different CSL (sub-)graphs is $\ell!/\ell^\ell$. Finally, since $\lim_{\ell \to \infty} \ell!/\ell^\ell = 0$, the probability goes to zero as $\ell$ grows. $\square$

Importantly, we derive the expected number of subgraphs that a random policy has to draw before identification.

**Lemma 6.** *Let $G$ be any graph in the family of non-isomorphic $(n, \ell)$-CSL graphs, namely $\mathcal{G}_{n,\ell}$ (Definition 2). The expected number of subgraphs that a random policy must uniformly draw before identification is $T = \Theta(\ell \ln \ell)$.*

*Proof.* Note that we have an instance of the coupon collector problem. The proof follows the steps in Mitzenmacher & Upfal (2017); Blom et al. (1993). Let $t_i$ be the time to collect (any node from) the $i$-th CSL (sub-)graphs, after $i - 1$ CSL (sub-)graphs have already been collected by drawing at least one of their nodes for marking. Let $p_i$ be probability of collecting a *new* CSL (sub-)graph (one from which nodes where not drawn before). Then,

$$p_i = \frac{\ell - (i-1)}{\ell} = \frac{\ell - i + 1}{\ell}.$$

Therefore, $t_i$ has geometric distribution with expectation $\mathbb{E}[t_i] = \frac{1}{p_i} = \frac{\ell}{\ell-i+1}$. By linearity of expectation, then the number of draws needed to collect all CSL (sub-)graphs is:

$$\begin{aligned}
\mathbb{E}[t_1 + t_2 + \cdots + t_\ell] =& \mathbb{E}[t_1] + \mathbb{E}[t_2] \ldots + \mathbb{E}[t_\ell] \\
=& \frac{1}{p_1} + \frac{1}{p_2} + \ldots + \frac{1}{p_\ell} \\
=& \frac{\ell}{\ell} + \frac{\ell}{\ell-1} + \ldots + \frac{\ell}{1} \\
=& \ell\Big(\frac{1}{1} + \frac{1}{2} + \ldots + \frac{1}{\ell}\Big) \\
=& \ell H_\ell
\end{aligned}$$

where $H_\ell$ is the $\ell$ harmonic number, which asymptotically grows as $\Theta(\ln \ell)$. Thus, we obtain the expectation $\Theta(\ell \ln \ell)$. $\qquad\square$

Our next theorem shows that POLICY-LEARN does not suffer from the shortcomings of the random selection. Indeed, POLICY-LEARN can provably implement the $\pi$ in Proposition 2, which in turn implies that it can learn to identify the isomorphism type of any graph in $\mathcal{G}_{n,\ell}$.

**Theorem 4** (POLICY-LEARN can identify $\mathcal{G}_{n,\ell}$). *Let $G$ be any graph in the family of non-isomorphic $(n,\ell)$-CSL graphs, namely $\mathcal{G}_{n,\ell}$ (Definition 2). There exist weights of $f$ and $g$ such that* POLICY-LEARN *with $T = \ell$ can identify the isomorphism type of $G$ within the family.*

*Proof.* We will show that POLICY-LEARN can implement the subgraph selection policy $\pi$ which returns $\ell$ subgraphs such that: (i) each subgraph is obtained by marking exactly one node and (ii) each marked node belongs to a different CSL (sub-)graph. Since $\pi$ is sufficient for identification (Proposition 2), then POLICY-LEARN can provably identify the isomorphism type of $G$.

Recall that we initialize the bag of subgraphs as $\mathcal{B}_G^0 := \{\!\{G\}\!\}$. At step $t = 0$, since all nodes are WL-indistinguishable (see proof of Theorem 1), then $p_1 = f(\mathcal{B}_G^0)$ is a uniform distribution. Thus, $v_1 \sim p_1$ is randomly sampled from the $\ell \cdot n$ possible nodes, and the bag is updated as $\mathcal{B}_G^1 = \mathcal{B}_G^0 \cup \{\!\{S_{v_1}\}\!\} = \{\!\{G, S_{v_1}\}\!\}$. At step $t = 1$, we obtain a distribution over the nodes as $p_2 = f(\mathcal{B}_G^1)$. Assuming $f$ has enough layers (it is sufficient[4] that the total number of layers $L_1$ is $n$), then all nodes belonging to the CSL (sub-)graph of node $v_1$ will have a different color than the remaining $(\ell - 1) \cdot n$ nodes. Therefore the MLP in Equation (3) can map them to have zero probability. On the contrary, the remaining $(\ell - 1) \cdot n$ nodes will all have the same color, and the MLP will map them to have the same probability. Thus $p_2$ will be uniform over all the nodes in the CSL (sub-)graphs other than the one containing node $v_1$. Therefore, $v_2 \sim p_2$ is randomly sampled from a CSL (sub-)graph different than one containing $v_1$. By repeating the argument, we have that at each iteration POLICY-LEARN samples one node from a CSL (sub-)graph that has not yet been selected. Thus, at $t = \ell$ the bag of subgraphs $\mathcal{B}_G^t$ will contain all the subgraphs that are sufficient for identification. $\qquad\square$

On the contrary, since OSAN (Qian et al., 2022) uses an MPNN as a selection network and samples all subgraphs at once, then it cannot perform better than randomly selecting subgraphs for the graphs in $\mathcal{G}_{n,\ell}$.

**Theorem 5** (Qian et al. (2022) cannot efficiently identify $\mathcal{G}_{n,\ell}$). *Let $G$ be any graph in the family of non-isomorphic $(n,\ell)$-CSL graphs, namely $\mathcal{G}_{n,\ell}$ (Definition 2). Then, the probability that OSAN (Qian et al., 2022) with $T = \ell$ subgraphs can successfully identify the isomorphism type of $G$ is $\ell!/\ell^\ell$, which tends to 0 as $\ell$ increases.*

*Proof.* Recall that the selection network in OSAN is parameterized as an MPNN taking as input the original graph and returning as output $\ell$ probability distributions over the $\ell \cdot n$ nodes, one for each of the $\ell$ subgraphs to sample. Since all nodes in $G$ are WL-indistinguishable (see proof of Theorem 1), then each of the $\ell$ probability distribution is uniform over the $\ell \cdot n$ nodes. Therefore, each node is sampled uniformly at random. Since the sampled nodes must belong to different CSL (sub-)graphs

---

[4]But not necessary, as for example the bound can be tightened to $n/2$ by considering the circular structure of the connected components, and even further by taking into account the skip connections.

for identification (Proposition 2), and because the probability of sampling $\ell$ nodes each belonging to a different CSL (sub-)graph from the uniform distribution is $\ell!/\ell^\ell$ (Proposition 3), then the probability that OSAN can identify the isomorphism type of $G$ is $\ell!/\ell^\ell$. $\qquad\square$

Finally, we prove that POLICY-LEARN can return all and only subgraphs having as root a node with degree greater than a predefined number.

**Proposition 7** (POLICY-LEARN can implement degree-aware policies). *Consider a policy $\pi$ that returns all $m$ node-marked subgraphs whose root nodes have degree greater than a predefined $d$, with $d \geq 1$. Then, POLICY-LEARN can implement $\pi$ with $T = m$.*

*Proof.* We will prove that we can implement degree-aware policies by showing that it exists a set of weights such that POLICY-LEARN gives a uniform probability to all nodes with degree $d$ that have not been already selected. Since the selection network is iteratively applied for $T = m$ times, then POLICY-LEARN would have selected all $m$ subgraphs after the last iteration.

Recall that we initialize the bag of subgraphs as $\mathcal{B}_G^0 := \{\!\{G\}\!\}$, and, at any step, we sample a node and add the corresponding subgraph to the bag. In the following, we drop the dependency on both the graph and the step in the notation, and call the current bag simply by $\mathcal{B}$.

Since we are only interested in sampling subgraphs having root nodes with degree $d$, we do not assume the presence of any additional initial node feature, and we directly work with a DS-GNN architecture with GraphConv encoders (Morris et al., 2019).[5] The representation for node $v$ in subgraph $i$ given by the $(l+1)$-th layer of our selection network can be written as:

$$h_{v,i}^{(l+1)} = \sigma\big(W_1^{(l)} h_{v,i}^{(l)} + W_2^{(l)} \sum_{u \sim_i v} h_{u,i}^{(l)} + b^{(l)}\big), \tag{5}$$

where $h_{v,i}^{(0)} := \big(\begin{smallmatrix} 1 \\ \mathrm{id}_{v,i} \end{smallmatrix}\big)$ with $\mathrm{id}_{v,i} = 0$ if $v$ is not the root of subgraph $i$, and $\mathrm{id}_{v,i} = 1$ if $v$ is the root of subgraph $i$. Recall that, given the representations for each node in each subgraph in the bag, we then obtain a distribution over the nodes in the original graph by simply pooling the final node representations $h_{v,i}^{(L)}$ at the final layer $L$, across all the subgraphs, i.e., the unnormalized node distribution is obtained as:

$$h_v = \mathrm{pool}_i(h_{v,i}^{(L)}). \tag{6}$$

We consider a selection Subgraph GNN with 4 layers (i.e., $L = 4$), and ReLU non-linearities. Each layer will have two channels, one used to compute an indicator variable storing whether a node has degree greater than $d$, and the other one propagating whether a node is the root of the subgraph. In the following we describe each layer in detail.

**First layer.** We set $W_1^{(0)} = \big(\begin{smallmatrix} 0 & 0 \\ 0 & 1 \end{smallmatrix}\big)$, $W_2^{(0)} = \big(\begin{smallmatrix} 1 & 0 \\ 0 & 0 \end{smallmatrix}\big)$, $b^{(0)} = \big(\begin{smallmatrix} -d \\ 0 \end{smallmatrix}\big)$, then for node $v$ we have,

$$h_{v,i}^{(1)} := \begin{pmatrix} h_{v,i}^{(1),0} \\ h_{v,i}^{(1),1} \end{pmatrix} = \sigma\left( \begin{pmatrix} 0 & 0 \\ 0 & 1 \end{pmatrix} \begin{pmatrix} 1 \\ \mathrm{id}_{v,i} \end{pmatrix} + \begin{pmatrix} 1 & 0 \\ 0 & 0 \end{pmatrix} \sum_{u \sim_i v} \begin{pmatrix} 1 \\ \mathrm{id}_{u,i} \end{pmatrix} + \begin{pmatrix} -d \\ 0 \end{pmatrix} \right)$$

$$= \sigma\left( \begin{pmatrix} d_i(v) - d \\ \mathrm{id}_{v,i} \end{pmatrix} \right)$$

where $d_i(v)$ is the degree of $v$ in subgraph $i$, and since we are using a node marked policy, it is equal to the degree of $v$ in $G$, namely $d(v)$. Note that in this way the zero-th entry of $h_{v,i}^{(1)}$, namely $h_{v,i}^{(1),0}$, is greater than 0 if and only if $d(v) > d$, and 0 otherwise.

---

[5]The extension to the presence of additional node features, as well as the usage of the separate propagation of them as in Dwivedi et al. (2021) is straightforward, and can be done by considering those additional weights as zeroed out.

**Second layer.** We set $W_1^{(1)} = \left(\begin{smallmatrix} -1 & 0 \\ 0 & 1 \end{smallmatrix}\right)$, $W_2^{(1)} = \left(\begin{smallmatrix} 0 & 0 \\ 0 & 0 \end{smallmatrix}\right)$, $b^{(1)} = \left(\begin{smallmatrix} 1 \\ 0 \end{smallmatrix}\right)$, then for node $v$ we have,

$$h_{v,i}^{(2)} := \begin{pmatrix} h_{v,i}^{(2),0} \\ h_{v,i}^{(2),1} \end{pmatrix} = \sigma\left(\begin{pmatrix} -1 & 0 \\ 0 & 1 \end{pmatrix} h_{v,i}^{(1)} + \begin{pmatrix} 0 & 0 \\ 0 & 0 \end{pmatrix} \sum_{u \sim_i v} h_{u,i}^{(1)} + \begin{pmatrix} 1 \\ 0 \end{pmatrix}\right)$$

$$= \sigma\left(\begin{pmatrix} -h_{v,i}^{(1),0} + 1 \\ h_{v,i}^{(1),1} \end{pmatrix}\right)$$

$$= \sigma\left(\begin{pmatrix} -h_{v,i}^{(1),0} + 1 \\ \mathrm{id}_{v,i} \end{pmatrix}\right)$$

Note that in this way the zero-th entry of $h_{v,i}^{(2)}$, namely $h_{v,i}^{(2),0}$, is 0 if $d(v) > d$, and it is equal to 1 if $d(v) \leq d$.

**Third layer.** We set $W_1^{(2)} = \left(\begin{smallmatrix} -1 & 0 \\ 0 & 1 \end{smallmatrix}\right)$, $W_2^{(2)} = \left(\begin{smallmatrix} 0 & 0 \\ 0 & 0 \end{smallmatrix}\right)$, $b^{(2)} = \left(\begin{smallmatrix} 1 \\ 0 \end{smallmatrix}\right)$, then for node $v$ we have,

$$h_{v,i}^{(3)} := \begin{pmatrix} h_{v,i}^{(3),0} \\ h_{v,i}^{(3),1} \end{pmatrix} = \sigma\left(\begin{pmatrix} -1 & 0 \\ 0 & 1 \end{pmatrix} h_{v,i}^{(2)} + \begin{pmatrix} 0 & 0 \\ 0 & 0 \end{pmatrix} \sum_{u \sim_i v} h_{u,i}^{(2)} + \begin{pmatrix} 1 \\ 0 \end{pmatrix}\right)$$

$$= \sigma\left(\begin{pmatrix} -h_{v,i}^{(2),0} + 1 \\ h_{v,i}^{(2),1} \end{pmatrix}\right)$$

$$= \sigma\left(\begin{pmatrix} -h_{v,i}^{(2),0} + 1 \\ \mathrm{id}_{v,i} \end{pmatrix}\right)$$

Note that in this way the zero-th entry of $h_{v,i}^{(3)}$, namely $h_{v,i}^{(3),0}$, is 0 if $d(v) \leq d$, and it is equal to 1 if $d(v) > d$. This means that at this point, $h_{v,i}^{(3),0}$ contains the binary information indicating whether node $v$ has degree greater than $d$. Therefore, we can use this entry to construct our distributions over nodes, ensuring we sample only nodes that have degree greater than $d$. However, we want to avoid re-sampling nodes that have already been sampled. We will show how this can be done by first using one additional layer that makes use of $\mathrm{id}_{v,i}$, $\forall v \in G$, and $\forall i \in \mathcal{B}$, and then relying on our pooling function.

**Fourth layer.** We set $W_1^{(3)} = \left(\begin{smallmatrix} 1 & -1 \end{smallmatrix}\right)$, $W_2^{(3)} = \left(\begin{smallmatrix} 0 & 0 \end{smallmatrix}\right)$, $b^{(3)} = 0$, then for node $v$ we have,

$$h_{v,i}^{(4)} = \sigma((1 \quad -1) h_{v,i}^{(3)} + (0 \quad 0) \sum_{u \sim_i v} h_{u,i}^{(3)} + 0)$$

$$= \sigma(h_{v,i}^{(3),0} - h_{v,i}^{(3),1})$$

$$= \sigma(h_{v,i}^{(3),0} - \mathrm{id}_{v,i})$$

Note that in this way $h_{v,i}^{(4)}$ is 1 if and only if $d(v) > d$ *and* $\mathrm{id}_{v,i} = 0$. Therefore, if we consider all the subgraphs in the current bag, if $v$ is the root of one subgraph in the bag, then there exist $i \in \mathcal{B}$ such that $h_{v,i}^{(4)} = 0$. In other words, if $\forall i \in \mathcal{B}$, we have $h_{v,i}^{(4)} = 1$, then it means that $d(v) > d$ and $v$ has not been already selected. On the contrary, if $\exists i \in \mathcal{B}$, such that $h_{v,i}^{(4)} = 0$, then it means that either $d(v) \leq d$ or $v$ has been already selected as root. Thus, we can rely on our pooling operation to create the node distribution.

**Pooling layer.** We set $\mathrm{pool}_i = \min_i$, and therefore

$$h_v = \min_{i \in \mathcal{B}}(h_{v,i}^{(4)}). \tag{7}$$

Note that if $h_v = 0$, then either $v$ has degree less than or equal to $d$ or it has been already selected as a root node (i.e., its corresponding subgraph is in $\mathcal{B}$). If instead $h_v = 1$, then $v$ has degree greater than $d$ and has not been selected. Thus, $h_v$ represents the unnormalised probability distribution over the nodes. $\square$

# D  ADDITIONAL EXPRESSIVE POWER ANALYSIS

In this section we extend the expressive power analysis reported in the main paper. We start by thoroughly comparing the expressive power of POLICY-LEARN and OSAN (Qian et al., 2022). While we have seen in Theorems 4 and 5 that there exist families of graphs that POLICY-LEARN can efficiently identify while OSAN cannot, we show here that the contrary is not true: there are no families of graphs that OSAN can identify and POLICY-LEARN cannot. This follows from the fact that on graphs of a fixed size $n$, for any instantiation of OSAN there exists a set of weights for POLICY-LEARN such that it outputs exactly the same probability distribution.

**Proposition 8** (POLICY-LEARN can implement OSAN). *Let $n$ be a natural number and consider all graphs of size $n$. Then there exists a set of weights for* POLICY-LEARN *such that it can parameterize all the probability distributions that OSAN can, but the contrary is not true.*

*Proof.* We start by showing that on all graphs of size $n$ POLICY-LEARN can parameterize all probability distributions that OSAN can.

Recall that the selection network in OSAN is parameterized as an MPNN taking as input the original graph and returning as output $T$ probability distributions over the $n$ nodes, one for each of the $T$ subgraphs to sample (simultaneously). This means that the output of the selection network in OSAN can be seen as a matrix of size $n \times T$. Consider now the selection network $f$ in POLICY-LEARN. We will show that there exists a set of weights such that at every iteration $t$ with $t \in \{1, \ldots, T\}$, $f$ outputs the same probability distribution that OSAN outputs for the $t$-th subgraph.

Let $t$ be the current iteration, and recall that $f$ outputs node features for every subgraph in the bag (Equation (2)), which are then pooled and passed through an MLP (Equation (3)). Note that if $f$ discards the marking information, then it implies that $f$ outputs the same node features for every subgraph in the bag, and can be parameterized exactly as the MPNN used in OSAN. Consider therefore the case where $f$ discards any marking information, and outputs for each subgraph a matrix of size $n \times (T + 1)$, where the first $T$ columns are exactly equal to the ones in the matrix returned by OSAN, and the last column is simply all ones. Therefore, if we pool the node representations across subgraphs using sum pooling, then the output is a matrix of size $n \times (T + 1)$, where the first $T$ columns are a multiple of the ones in OSAN, and the last column contains the number of subgraphs in the bag, or, equivalently, the iteration $t$. Finally, we rely on the ability of MLPs to memorize a finite number of input-output pairs (see Yun et al. (2019) and Yehudai et al. (2021, Lemma B.2)). We use that result to show that there exists a point-wise MLP that transforms, for each row, the value $t$ in the last column into its one-hot representation, thus obtaining a matrix $n \times (T + T)$, where once again the first $T$ columns are identical to the ones in OSAN and the last $T$ columns are used to maintain for each row a one-hot representation of $t$. Finally, we only need a final point-wise MLP that computes, for each row, the inner product between the first $T$ entries and the last $T$ entries. Since we have a finite number of graphs, we only need to make sure that the MLP memorizes a finite number of such inner products, and the lemma mentioned above in Yehudai et al. (2021) guarantees there is such an MLP. This MLP effectively implements a selection of the $t$-th column of the matrix using the one-hot representation of $t$ encoded in the last columns, and the $t$-th column corresponds to the distribution returned by OSAN for the $t$ subgraph. This means that for every iteration $t$ POLICY-LEARN can output the probability distribution returned by OSAN for the $t$-th subgraph.

Furthermore, from Theorems 4 and 5 it follows that OSAN cannot parameterize all probability distributions that POLICY-LEARN can. □

While the above result compared the expressive power of OSAN and POLICY-LEARN, it is also important to understand how well POLICY-LEARN compares to the full-bag approach on different families of graphs. In Section 4 we have already discussed that for CSL graphs it is sufficient to mark any single node to obtain the same expressive power of the full-bag. However, this result can be extended to other non-isomorphic WL-indistinguishable families of graphs. Specifically, we show next that marking any node is sufficient to distinguish strongly regular graphs of different parameters $(n, k, \lambda, \mu)$ (i.e., number of nodes, degree, number of common neighbors with adjacent and non-adjacent nodes), but, just like 3-WL and node-based full-bag Subgraph GNNs, POLICY-LEARN cannot distinguish strongly regular graphs of the same parameters.

**Proposition 9** (POLICY-LEARN can distinguish certain strongly-regular graphs). POLICY-LEARN *can distinguish any strongly regular graphs of different parameters, but cannot distinguish any strongly regular graphs of the same parameters.*

*Proof.* The proof extends Bevilacqua et al. (2022, Lemma 14) to the node-marking generation policy, and follows the same steps. We show that the WL algorithm converges to the *same* coloring of any subgraph obtained by marking a single node of a strongly regular graph, and this coloring depends only on the parameters of the strongly regular graph. Let $G$ be a strongly regular graph with parameters $(n, k, \lambda, \mu)$, i.e., $G$ has $n$ nodes, is regular of degree $k$, any two adjacent nodes have $\lambda$ common neighbors, any two non-adjacent nodes have $\mu$ common neighbors. Consider the subgraph obtained by marking node $u$. We initialize all nodes to the same color 1, except node $u$ which has color 2.

(Iter. 1)  Nodes adjacent to $u$ are colored 3, nodes non-adjacent to $u$ are colored 4, $u$ is colored 5.

(Iter. 2)  Nodes adjacent to $u$ are colored 6, as they each had: (i) color 3; (ii) $\lambda$ neighbors of color 3; (iii) $(k - 1 - \lambda)$ neighbors of color 4; (iv) one neighbor of color 5. To see this, recall that these nodes are adjacent to $u$, so they have $\lambda$ common neighbors with $u$, and these nodes had color 3 at iteration 1 because they are indeed also neighbors with $u$. This also implies that the remaining $(k - 1 - \lambda)$ neighbors necessarily had color 4 at iteration 2.

Nodes non-adjacent to $u$ are colored 7, as they each had: (i) color 4; (ii) $\mu$ neighbors of color 3; (iii) $(k - \mu)$ neighbors of color 4. This is because they are non-adjacent to $u$, so they have exactly $\mu$ common neighbors with $u$, and these nodes had color 3 at iteration 2. Similarly, the remaining $(k - \mu)$ neighbors had color 4 at iteration 2.

Node $u$ is colored 8 as it had: (i) color 5; (ii) $k$ neighbors of color 3.

Since the colors at iteration 2 are isomorphic to the colors at iteration 1, WL converged. Since this coloring is the same for any node-marked subgraph of any strongly regular graph of the same parameters, then POLICY-LEARN cannot distinguish strongly regular graph of the same parameters. Similarly, the coloring differs for strongly regular graphs of different parameters: at initialization if $n$ differs, at iteration 1 if $k$ differs, at iteration 2 if $\lambda$ or $\mu$ differs. Thus POLICY-LEARN can distinguish strongly regular graphs of different parameters.  □

Proposition 9 has important implications on the relation with higher order WL tests, as strongly regular graphs are instead distinguishable by 4-WL. Since there exist pairs of 4-WL distinguishable graphs that are not distinguishable with our approach, then POLICY-LEARN is not as powerful as 4-WL.

Finally, we remark here that it is also possible to consider other families of graphs, beyond the family of non-isomorphic $(n, \ell)$-CSL graphs (Definition 2), where marking a limited number of nodes is sufficient for identifiability, and therefore where POLICY-LEARN can attain the same expressive power of the full bag while using a significantly reduced set of subgraphs. Indeed, our theoretical results are valid for any family of graphs, as we describe next, obtained from any collection of WL-indistinguishable graphs that become distinguishable when marking any node, e.g., strongly regular graphs of different parameters. Similarly to the construction of the $(n, \ell)$-CSL family, each graph in these families is created by considering $\ell$ disconnected, non-isomorphic instances in the corresponding collection.

# E  ADDITIONAL EXPERIMENTS AND DETAILS

## E.1  ADDITIONAL RESULTS ON THE OGB, ZINC AND REDDIT-BINARY DATASETS

We additionally compared POLICY-LEARN to a wide range of baselines, including existing full-bag Subgraph GNNs, on the OGB, ZINC and REDDIT-BINARY datasets presented in the main paper. We further report the random baseline using the prediction network considered in OSAN, whose results are taken from Qian et al. (2022) and which we denote as RANDOM-OSAN. The results on the various OGB datasets are presented in Table 4. Interestingly, POLICY-LEARN consistently

Table 4: Comparison of POLICY-LEARN to standard MPNNs and full-bag Subgraph GNNs demonstrate the advantages of our approach. $-$ indicates the results was not reported in the original paper.

| Method ↓ / Dataset → | MOLESOL RMSE ↓ | MOLTOX21 ROC-AUC ↑ | MOLBACE ROC-AUC ↑ | MOLHIV ROC-AUC ↑ |
|---|---|---|---|---|
| **MPNNs** | | | | |
| GCN (Kipf & Welling, 2017) | 1.114±0.036 | 75.29±0.69 | 79.15±1.44 | 76.06±0.97 |
| GIN (Xu et al., 2019) | 1.173±0.057 | 74.91±0.51 | 72.97±4.00 | 75.58±1.40 |
| **FULL-BAG SUBGRAPH GNNs** | | | | |
| RECONSTR. GNN (Cotta et al., 2021) | 1.026±0.033 | 75.15±1.40 | $-$ | 76.32±1.40 |
| NGNN (Zhang & Li, 2021) | $-$ | $-$ | $-$ | 78.34±1.86 |
| DS-GNN (EGO+) (Bevilacqua et al., 2022) | $-$ | 76.39±1.18 | $-$ | 77.40±2.19 |
| DSS-GNN (EGO+) (Bevilacqua et al., 2022) | $-$ | 77.95±0.40 | $-$ | 76.78±1.66 |
| GNN-AK+ (Zhao et al., 2022) | $-$ | $-$ | $-$ | 79.61±1.19 |
| SUN (EGO+) (Frasca et al., 2022) | $-$ | $-$ | $-$ | 80.03±0.55 |
| GNN-SSWL+ (Zhang et al., 2023) | $-$ | $-$ | $-$ | 79.58±0.35 |
| FULL | 0.847±0.015 | 76.25±1.12 | 78.41±1.94 | 76.54±1.37 |
| **SAMPLING SUBGRAPH GNNs** | | | | |
| RANDOM-OSAN (Qian et al., 2022) $T = 10$ | 1.128±0.055 | $-$ | 71.90±3.90 | $-$ |
| OSAN (Qian et al., 2022) $T = 10$ | 0.984±0.086 | $-$ | 72.30±6.60 | $-$ |
| RANDOM $T = 2$ | 0.951±0.039 | 76.65±0.89 | 75.36±4.28 | 77.55±1.24 |
| POLICY-LEARN $T = 2$ | 0.877±0.029 | 77.47±0.82 | 78.40±2.85 | 79.13±0.60 |
| RANDOM $T = 5$ | 0.900±0.032 | 76.62±0.63 | 78.14±2.36 | 77.30±2.56 |
| POLICY-LEARN $T = 5$ | 0.883±0.032 | 77.36±0.60 | 78.39±2.28 | 78.49±1.01 |

approaches or even outperforms computationally-intensive full-bag methods across these datasets. On the ZINC-12K dataset (Table 5), POLICY-LEARN with $T = 5$ performs similarly to NGNN (Zhang & Li, 2021), which uses on average $5\times$ more subgraphs. Additional comparisons with other values of $T$, namely $T = 3$ and $T = 8$, align with the observations made for $T = 2$ and $T = 5$. In particular, POLICY-LEARN always outperforms the random baseline, and the gap is larger when the number of subgraphs $T$ is smaller, where selecting the most informative subgraphs is more crucial. Finally, results on the REDDIT-BINARY dataset (Table 6) demonstrate the applicability of our method to cases where full-bag Subgraph GNNs, such as our implementation FULL, as well as DS-GNN and DSS-GNN (Bevilacqua et al., 2022), are otherwise inapplicable. Notably POLICY-LEARN surpasses the random counterparts that uniformly sample the same number of subgraphs, even when coupled with the DS-GNN or DSS-GNN downstream prediction model proposed in Bevilacqua et al. (2022).

## E.2 TIME COMPARISONS

We performed timing analysis on REDDIT-BINARY and ZINC datasets, which we discuss in the following.

**Timings on REDDIT-BINARY.** To further underscore the effectiveness of POLICY-LEARN on the REDDIT-BINARY dataset, we conducted a time comparison, presented in Table 7. Specifically, we estimated the inference time on the entire test set using a batch size of 128 on an NVIDIA RTX A6000 GPU. We measure the times in two scenarios: when setting `torch.use_deterministic_algorithms(True)` and without setting it (which is equivalent to setting it to `False`). While the former ensures deterministic results, it comes at the cost of an increased running time, and it is not usually set in Subgraph GNNs implementations. Nonetheless, since determinism might be important in practice, we report both results. We compared POLICY-LEARN with the RANDOM baseline, maintaining the same hyperparameters for all methods for a fair comparison. As expected, POLICY-LEARN takes longer than RANDOM with $T = 2$, but also obtains better results (see Table 6). To further verify its effectiveness, we ran RANDOM with $T = 20$, and observed that it takes more than twice the time of POLICY-LEARN (and gets worse performance, see Table 6), showcasing how our approach is beneficial in practice.

**Timings on ZINC-12K.** We report empirical runtimes on the ZINC dataset in Table 8. For all methods, we estimated the training time on the entire training set as well as the inference time on the entire test set using a batch size of 128 on an NVIDIA RTX A6000 GPU. For all these experiments,

Table 5: Comparison of POLICY-LEARN to existing expressive methods on the ZINC-12K graph dataset. All methods obey to the 500k parameter budget.

| Method | ZINC (MAE ↓) |
|---|---|
| **MPNNs** | |
| GCN (Kipf & Welling, 2017) | 0.321±0.009 |
| GIN (Xu et al., 2019) | 0.163±0.004 |
| **EXPRESSIVE GNNs** | |
| PNA (Corso et al., 2020) | 0.133±0.011 |
| GSN (Bouritsas et al., 2022) | 0.101±0.010 |
| CIN (Bodnar et al., 2021a) | 0.079±0.006 |
| **FULL-BAG SUBGRAPH GNNs** | |
| NGNN (Zhang & Li, 2021) | 0.111±0.003 |
| DS-GNN (EGO+) (Bevilacqua et al., 2022) | 0.105±0.003 |
| DSS-GNN (EGO+) (Bevilacqua et al., 2022) | 0.097±0.006 |
| GNN-AK (Zhao et al., 2022) | 0.105±0.010 |
| GNN-AK+ (Zhao et al., 2022) | 0.091±0.011 |
| SUN (EGO+) (Frasca et al., 2022) | 0.084±0.002 |
| GNN-SSWL (Zhang et al., 2023) | 0.082±0.003 |
| GNN-SSWL+ (Zhang et al., 2023) | 0.070±0.005 |
| FULL | 0.087±0.003 |
| **SAMPLING SUBGRAPH GNNs** | |
| RANDOM-OSAN (Qian et al., 2022) $T = 2$ | 0.214±0.007 |
| OSAN (Qian et al., 2022) $T = 2$ | 0.177±0.016 |
| RANDOM $T = 2$ | 0.136±0.005 |
| POLICY-LEARN $T = 2$ | 0.120±0.003 |
| RANDOM $T = 3$ | 0.128±0.004 |
| POLICY-LEARN $T = 3$ | 0.116±0.008 |
| RANDOM $T = 5$ | 0.113±0.006 |
| POLICY-LEARN $T = 5$ | 0.109±0.005 |
| RANDOM $T = 8$ | 0.102±0.003 |
| POLICY-LEARN $T = 8$ | 0.097±0.005 |

Table 6: Results on the REDDIT-BINARY dataset. POLICY-LEARN outperforms all baselines, including Subgraph GNNs coupled with random selection policies.

| Method | RDT-B (ACC ↑) |
|---|---|
| GIN (Xu et al., 2019) | 92.4±2.5 |
| SIN (Bodnar et al., 2021b) | 92.2±1.0 |
| CIN (Bodnar et al., 2021a) | 92.4±2.1 |
| RANDOM DS-GNN (Bevilacqua et al., 2022) $T = 2$ | 91.3±1.6 |
| RANDOM DSS-GNN (Bevilacqua et al., 2022) $T = 2$ | 92.7±0.8 |
| FULL | OOM |
| RANDOM $T = 20$ | 92.6±1.5 |
| RANDOM $T = 2$ | 92.4±1.0 |
| POLICY-LEARN $T = 2$ | 93.0±0.9 |

we set `torch.use_deterministic_algorithms(False)`. During training, the runtime of POLICY-LEARN is significantly closer to that of the RANDOM approach than to the one of the full-bag Subgraph GNN FULL, and POLICY-LEARN is faster than OSAN. At inference, POLICY-LEARN places in between RANDOM and FULL, and it is significantly faster than OSAN while also achieving better results.

Finally, we compare the time and the prediction performance on the ZINC dataset with the CIN model (Bodnar et al., 2021a) in Table 9. For all methods, we report the inference time on the entire test set using a batch size of 128. The runtime of CIN is taken from the original paper, and it is measured on an NVIDIA Tesla V100 GPU. To ensure a fair comparison, we therefore timed POLICY-LEARN on

Table 7: Timing comparison on the REDDIT-BINARY dataset on an RTX A6000 GPU (48 GB). Time taken at inference on the test set. Results on the left column are obtained when setting `torch.use_deterministic_algorithms(True)`, while those on the right column are without it. All values are in milliseconds.

| Method | | RDT-B (Time (ms)) | |
|--------|--------|--------|--------|
| | | det=True | det=False |
| FULL | | OOM | OOM |
| RANDOM | $T = 20$ | $1360.0\pm1.3$ | $180.65\pm5.2$ |
| RANDOM | $T = 2$ | $216.7\pm0.4$ | $39.7\pm2.3$ |
| POLICY-LEARN | $T = 2$ | $411.7\pm0.6$ | $77.4\pm1.2$ |

Table 8: Timing comparison on the ZINC-12K dataset on an RTX A6000 GPU (48 GB). Time taken at train for one epoch and at inference on the test set. All values are in milliseconds.

| Method | | ZINC | | |
|--------|--------|--------|--------|--------|
| | | Train time (ms) | Test time (ms) | MAE $\downarrow$ |
| GIN (Xu et al., 2019) | | $1370.10\pm10.79$ | $84.81\pm0.26$ | $0.163\pm0.004$ |
| FULL | | $4872.79\pm14.30$ | $197.38\pm0.30$ | $0.087\pm0.003$ |
| OSAN (Qian et al., 2022) | $T = 2$ | $2964.46\pm30.36$ | $227.93\pm0.21$ | $0.177\pm0.016$ |
| RANDOM | $T = 2$ | $2114.00\pm27.88$ | $107.02\pm0.22$ | $0.136\pm0.005$ |
| POLICY-LEARN | $T = 2$ | $2489.25\pm\ 9.42$ | $150.38\pm0.33$ | $0.120\pm0.003$ |

the same GPU type. First, we observe that POLICY-LEARN is faster than CIN, especially considering the additional preprocessing time required by CIN for the graph lifting procedures, which is not measured in Table 9. Second, although CIN outperforms POLICY-LEARN, it should be noted that CIN explicitly models cycles and rings, which are obtained in the preprocessing step and serve as a domain-specific inductive bias. On the contrary, the subgraph generation policy in POLICY-LEARN (i.e., node-marking) is entirely domain-agnostic and not tailored to the specific application.

### E.3 ADDITIONAL DATASETS

**Alchemy.** We conducted an additional experiment considering the Alchemy-12K dataset (Chen et al., 2019), where we compared not only to the direct baselines FULL and RANDOM, but also to a positional and structural encoding augmented method GIN + RWSE & LAPPE, as reported in Qian et al. (2023). As can be seen from Table 10, these results further demonstrates the capabilities of POLICY-LEARN, which surpasses the random baseline RANDOM, and get results close to those obtained by the FULL approach.

**Expressivity Datasets.** In an effort to answer whether POLICY-LEARN maintains the expressive power of full-bag Subgraph GNNs in practice, we experimented on the CSL (Murphy et al., 2019; Dwivedi et al., 2020) and EXP (Abboud et al., 2020) synthetic datasets, which are constructed in a way that 1-WL GNNs cannot outperform the performance of a random guess (10% accuracy in CSL, 50% accuracy in EXP). We followed the evaluation procedure in Bevilacqua et al. (2022), consisting of a k-fold cross validation ($k = 5$ in CSL, $k = 10$ in EXP). Results are reported in Table 11. On the CSL dataset, both RANDOM and POLICY-LEARN achieve 100% test accuracy, aligning with our theoretical analysis at the beginning of Section 4, where we discussed the disambiguation of these graphs. On the EXP dataset, POLICY-LEARN surpasses the performance of RANDOM, and achieves the same performance of the full-bag approach. This numerically confirms the importance of learning which subgraphs to select, rather than randomly sampling them.

**Counting Datasets.** We tested the abilities of counting substructures on the benchmarks from Chen et al. (2020), on which full-bag Subgraph GNNs have been extensively evaluated (Zhao et al., 2022; Frasca et al., 2022). We used the dataset splits and evaluation procedure of Zhao et al. (2022); Frasca et al. (2022) and reported the average across the seeds in Table 12. While POLICY-LEARN

Table 9: Timing comparison with CIN (Bodnar et al., 2021a) on the ZINC-12K dataset on an RTX A100 GPU. Time taken at inference on the test set. All values are in milliseconds.

| Method | | ZINC | |
|---|---|---|---|
| | | Time (ms) | MAE ↓ |
| GIN (Xu et al., 2019) | | 126.91±0.82 | 0.163±0.004 |
| CIN (Bodnar et al., 2021a) | | 471.00±3.00 | 0.079±0.006 |
| POLICY-LEARN | $T = 2$ | 235.14±0.21 | 0.120±0.003 |
| POLICY-LEARN | $T = 5$ | 411.19±0.39 | 0.109±0.005 |

Table 10: Comparison of POLICY-LEARN to existing methods on the ALCHEMY-12K graph dataset.

| Method | ALCHEMY (MAE ↓) |
|---|---|
| **MPNNs** | |
| GIN (Xu et al., 2019) | 11.12±0.690 |
| GIN + RWSE & LAPPE (Dwivedi et al., 2020) | 7.197±0.094 |
| **FULL-BAG SUBGRAPH GNNs** | |
| FULL | 6.65±0.143 |
| **SAMPLING SUBGRAPH GNNs** | |
| RANDOM-OSAN (Qian et al., 2022) $T = 10$ | 12.11±0.210 |
| OSAN (Qian et al., 2022) $T = 10$ | 8.87±0.120 |
| RANDOM $T = 2$ | 6.95±0.148 |
| POLICY-LEARN $T = 2$ | 6.89±0.151 |
| RANDOM $T = 5$ | 6.78±0.174 |
| POLICY-LEARN $T = 5$ | 6.72±0.107 |

Table 11: Results on the CSL and EXP expressivity datasets. POLICY-LEARN exhibits the same disambiguation capabilities of the full-bag.

| Method | CSL (ACC ↑) | EXP (ACC ↑) |
|---|---|---|
| GIN (Xu et al., 2019) | 10.00±0.0 | 51.20±2.1 |
| FULL | 100.00±0.0 | 100.00±0.0 |
| RANDOM $T = 2$ | 100.00±0.0 | 89.92±2.5 |
| POLICY-LEARN $T = 2$ | 100.00±0.0 | 100.00±0.0 |

cannot achieve the same performance of the full-bag, it significantly surpasses the random baseline. Therefore, although counting all substructures might not be possible with a significantly reduced set of subgraphs, there exist subgraph selections that enable more effective counting. Furthermore, increasing the number of subgraphs proves advantageous in terms of the counting power.

### E.4 EXPERIMENTAL DETAILS

We implemented POLICY-LEARN using Pytorch (Paszke et al., 2019) and Pytorch Geometric (Fey & Lenssen, 2019). We ran our experiments on NVIDIA DGX V100, GeForce 2080, NVIDIA RTX A5000, NVIDIA RTX A6000, NVIDIA GeForce RTX 4090 and TITAN V GPUs. We performed hyperparameter tuning using the Weight and Biases framework (Biewald, 2020). We used mean aggregator to aggregate node representations across subgraphs in the selection network $f$. Similarly, in the prediction network $g$, we use mean aggregator to obtain subgraph representations given the representations of nodes in each subgraph, and mean aggregator to obtain graph representations given the subgraph representations. We always employ GIN layers (Xu et al., 2019) to perform message passing, and make use of Batch Normalization, with disabled computed statistics in the selection model (as the number of subgraphs changes every $t$). Our MLPs are composed of two linear layers with ReLU non-linearities. Unless otherwise specified, we use residual connections. Each experiment is repeated for 5 different seeds. During evaluation, we replace sampling with the $\arg\max$ function and choose the node with the highest probability. Details of hyperparameter grid for each dataset can be found in the following subsections.

Table 12: Comparison of counting capabilities on the datasets in Chen et al. (2020). While reducing the set of subgraphs necessarily impacts the counting abilities of Subgraph GNNs, POLICY-LEARN significantly outperforms RANDOM.

| Method | Counting Substructures (MAE $\downarrow$) | | | |
|---|---|---|---|---|
| | Triangle | Tailed Tri. | Star | 4-Cycle |
| GIN (Xu et al., 2019) | 0.3569 | 0.2373 | 0.0224 | 0.2185 |
| FULL | 0.0183 | 0.0170 | 0.0106 | 0.0210 |
| RANDOM $T = 5$ | 0.1841 | 0.1259 | 0.0105 | 0.1180 |
| POLICY-LEARN $T = 5$ | 0.1658 | 0.1069 | 0.0100 | 0.0996 |
| RANDOM $T = 8$ | 0.1526 | 0.0999 | 0.0119 | 0.0933 |
| POLICY-LEARN $T = 8$ | 0.1349 | 0.0801 | 0.0100 | 0.0793 |

### E.4.1   OGB DATASETS

We considered the challenging scaffold splits proposed in Hu et al. (2020), and for each dataset we used the loss and evaluation metric prescribed therein. For all models (FULL, RANDOM, POLICY-LEARN), we used Adam optimizer with initial learning rate of $0.001$. We set the batch size to 128, except for the FULL method on MOLBACE and MOLTOX21 where we reduced it to 32 to avoid out-of-memory errors. We decay the learning rate by $0.5$ every 300 epochs for all dataset except MOLHIV, where we followed the choices of Frasca et al. (2022), namely constant learning rate, downstream prediction network with 2 layers, embedding dimension 64 and dropout in between layers with probability in $0.5$. We used $5$ layers with embedding dimension 300 and dropout in between layers with probability in $0.5$ in the prediction network for MOLTOX21 and MOLBACE as prescribed by Hu et al. (2020). For MOLESOL we employed a downstream prediction network with layers in $\{2, 3\}$ and embedding dimension 64. In all datasets, for POLICY-LEARN we employed a selection network architecturally identical to the prediction network, trained with a separate Adam optimizer without learning rate decay and learning rate $0.001$. We tuned the temperature parameter $\tau$ of the Gumbel-Softmax trick in $\{0.33, 0.66, 1, 2\}$. To prevent overconfidence in the probability distribution over nodes, we added a dropout during train, with probability tuned in $\{0, 0.3, 0.5\}$.

The maximum number of epochs is set to $1000$ for all models and datasets except MOLHIV and MOLTOX21, where it is reduced to 500. The test metric is computed at the best validation epoch.

### E.4.2   ZINC-12K

We considered the dataset splits proposed in Dwivedi et al. (2020), and used Mean Absolute Error (MAE) both as loss and evaluation metric. For all models (FULL, RANDOM, POLICY-LEARN), we used a batch size of 128 and Adam optimizer with initial learning rate of $0.001$, which is decayed by $0.5$ every 300 epochs. The maximum number of epochs is set to $1000$ for RANDOM and POLICY-LEARN when $T = 2$, and to 600 for RANDOM and POLICY-LEARN when $T = 5$, as well as for FULL, due to the increase in training time. The test metric is computed at the best validation epoch. We used a downstream prediction network composed of 6 layers, embedding dimension 64. For POLICY-LEARN, we employed a selection network architecturally identical to the prediction network, trained with a separate Adam optimizer without learning rate decay and learning rate $0.001$. We further tuned the temperature parameter $\tau$ of the Gumbel-Softmax trick in $\{0.33, 0.66, 1, 2\}$. Early experimentation revealed that POLICY-LEARN became overconfident in the probability of specific nodes, thus preventing the exploration of other nodes through sampling. We mitigated the problem by adding dropout on the node probability distribution during train, with probability tuned in $\{0, 0.3, 0.5\}$. For $T = 5$ we additionally masked out nodes corresponding to subgraphs that had already been selected, to avoid repetitions and encourage exploration especially at training time. To ensure a fair comparison, we applied a similar mechanism to RANDOM with $T = 5$, preventing it to sample the same subgraph (i.e., we did not allow replacement when uniformly sampling a subset of subgraphs from the bag). For the additional experiments featuring other values of $T$, namely $T = 3$ and $T = 8$, we use the same strategy employed for $T = 5$.

### E.4.3  REDDIT-BINARY

We used the evaluation procedure proposed in Xu et al. (2019), consisting of 10-fold cross validation and metric at the best averaged validation accuracy across the folds. The downstream prediction network is composed by 4 layers with embedding dimension 32 and no residual connections. We use Adam optimizer with learning rate tuned in $\{0.01, 0.001\}$. For the selection network we use the same architectural choices adopted for the prediction network, and the same learning rate and optimizer type. We consider a batch size of 128, and trained for 500 epochs. We encouraged exploration during train through dropout on the node probability distribution, with probability tuned in $\{0, 0.5\}$. Finally, we tuned the temperature $\tau$ within the Gumbel-Softmax in $\{0.33, 0.66, 1, 2\}$.

### E.4.4  ALCHEMY

We used the same dataset splits of Morris et al. (2020b), and adopted the evaluation procedure followed by Qian et al. (2022), which differs from Morris et al. (2020b) only in the additional final denormalization of the predictions and ground-truths. For all models (FULL, RANDOM, POLICY-LEARN), we used a batch size of 128 and Adam optimizer with initial learning rate of $0.001$, which is decayed by $0.5$ every 300 epochs. The maximum number of epochs is set to 1000, and the test metric is computed at the best validation metric. Following Morris et al. (2020b), we used a downstream prediction network composed of 6 layers and embedding dimension 64. For POLICY-LEARN, we employed a selection network architecturally identical to the prediction network, trained with a separate Adam optimizer without learning rate decay and learning rate $0.001$. We further tuned the temperature parameter $\tau$ of the Gumbel-Softmax trick in $\{0.33, 0.66, 1, 2\}$, and, to prevent overconfidence in the probability distribution over nodes, we added a dropout during train, with probability tuned in $\{0, 0.3, 0.5\}$. Similarly to ZINC, when $T = 5$ we mask out already selected subgraphs both for POLICY-LEARN and RANDOM, to avoid repeated subgraph selections.

### E.4.5  EXPRESSIVITY DATASETS

Following Bevilacqua et al. (2022), we used the same training procedure and evaluation strategy considered for the REDDIT-BINARY datasets. The downstream prediction network is composed by 6 layers with embedding dimension 64. For the EXP dataset, we additionally used residual connections and set to False the `track_running_stats` flag in the batch normalization, thus employing batch statistics instead of dataset ones during evaluation. We used Adam optimizer with learning rate tuned in $\{0.01, 0.001\}$ for CSL and set to $0.0001$ for EXP. For the selection network we used the same architectural choices adopted for the prediction network and the same optimizer type, with learning rate set to $0.001$ for CSL and $0.0001$ for EXP. We consider a batch size of 8 for CSL and 128 for EXP, and trained for 100 epochs. We fixed the temperature $\tau$ within the Gumbel-Softmax to 1. We mask out already selected subgraphs both for POLICY-LEARN and RANDOM, to avoid repeated subgraph selections.

### E.4.6  COUNTING DATASETS

We considered the dataset splits of previous work (Zhao et al., 2022; Frasca et al., 2022), and used Mean Absolute Error (MAE) both as loss and evaluation metric. For all models (FULL, RANDOM, POLICY-LEARN), we used a batch size of 128 and Adam optimizer with initial learning rate of $0.001$, which is decayed by $0.5$ every 300 epochs. The maximum number of epochs is set to 1000 and the test metric is measured at the best validation epoch. We used a downstream prediction network composed of 6 layers, embedding dimension 32. For POLICY-LEARN, we employed a selection network architecturally identical to the prediction network, trained with a separate Adam optimizer without learning rate decay and learning rate $0.001$. We further tuned the temperature parameter $\tau$ of the Gumbel-Softmax trick in $\{0.33, 0.66, 1, 2\}$, and, to prevent overconfidence in the probability distribution over nodes, we added a dropout during train, with probability tuned in $\{0, 0.3, 0.5\}$. For both POLICY-LEARN and RANDOM, we masked out already selected subgraphs, therefore preventing it to sample the same subgraph.

# F    COMPLEXITY ANALYSIS

In this section we analyze the complexity of POLICY-LEARN and compare it with its full-bag Subgraph GNN counterpart. We analyze here an equivalent efficient implementation of our method, which differs from Algorithm 1 in that it does not feed to the selection network the entire current bag at every iteration. Instead, it processes only the last subgraph added to the bag, and re-uses the representations of the previously selected subgraphs, passed though the selection network in previous iterations and stored. In what follows, we consider the feature dimensionality and the number of layers to be constants. We denote $\Delta_{\max}$ the maximum node degree of the input graph, $n$ the number of its nodes and $T$ the number of selected subgraphs.

**Time complexity.** The forward-pass asymptotic time complexity of the selection network $f$ amounts to $\mathcal{O}(T \cdot n \cdot \Delta_{\max})$. This arises from performing $T$ iterations, where each iteration involves selecting a new subgraph by passing the last subgraph added to the bag through the MPNN, which has time complexity $\mathcal{O}(n \cdot \Delta_{\max})$ and re-using stored representations of previously-selected subgraphs. The forward-pass asymptotic time complexity of the prediction network $g$ is $\mathcal{O}((T+1) \cdot n \cdot \Delta_{\max})$, as each of the $T + 1$ subgraphs is processed through the MPNN, which is different than the one used in the selection, in $\mathcal{O}(n \cdot \Delta_{\max})$. Note that, differently from the selection network, the prediction network considers $T + 1$ subgraphs as it also utilizes the last selected subgraph. Therefore, POLICY-LEARN has an overall time complexity of $\mathcal{O}((T + (T + 1)) \cdot n \cdot \Delta_{\max})$, i.e., $\mathcal{O}(T \cdot n \cdot \Delta_{\max})$.

**Space complexity.** The forward-pass asymptotic space complexity of the selection network $f$ is $\mathcal{O}(T \cdot n + (n + n \cdot \Delta_{\max}))$, because we need to store $n$ node features for each of the $T$ subgraphs, and, at each iteration, the space complexity of the MPNN on the subgraph being processed is $\mathcal{O}(n + n \cdot \Delta_{\max})$ to store its node features and connectivity in memory. Similarly, the forward-pass asymptotic space complexity of the prediction network $g$ is $\mathcal{O}((T + 1) \cdot (n + n \cdot \Delta_{\max}))$. Therefore, POLICY-LEARN has an overall space complexity of $\mathcal{O}(T \cdot (n + n \cdot \Delta_{\max}))$.

Notably, POLICY-LEARN is advantageous in both time and space complexity when compared to full-bag Subgraph GNNs. Indeed, as studied in Bevilacqua et al. (2022), the time complexity of full-bag node-based Subgraph GNNs amounts to $\mathcal{O}(n^2 \cdot \Delta_{\max})$, while the space complexity is $\mathcal{O}(n \cdot (n + n \cdot \Delta_{\max}))$, since the number of subgraphs in the full bag is exactly $n$. Since POLICY-LEARN scales linearly with $T$, when small values of $T$ are used, as in our paper, this results in a speedup by a factor of $n$ compared to full-bag Subgraph GNNs.

To grasp the impact of this reduction, let us consider for example the REDDIT-BINARY dataset, where the average number of nodes per graph is approximately $n = 429$, while the number of selected subgraphs is $T = 2$. The time and space complexities are drastically reduced, because $T \ll n$, and indeed POLICY-LEARN can be effectively run while the full bag Subgraph GNN is computationally infeasible.

