# OpenReview forum: "Efficient Subgraph GNNs by Learning Effective Selection Policies"
_ICLR.cc/2024/Conference — ICLR 2024 poster_

### Official Review · Reviewer_ew9j · 2023-10-16

**Soundness:** 3 good
**Presentation:** 3 good
**Contribution:** 2 fair
**Rating:** 6
**Confidence:** 4

**Summary:**

This paper is motivated by the high computational cost of subgraph GNNs and the bag-of-subgraph contains many redundant information and proposed a learnable method to efficiently select a fixed number of subgraphs for downstream prediction. The proposed method achieves a good balance between the cost and the expressive power. It obtains reasonable performance across various datasets.

**Strengths:**

1. The paper is overall sound and easy to follow.
2. The motivated CSL example is clear.
3. The proposed method achieves reasonable results across various datasets.

**Weaknesses:**

The main weaknesses of the paper lie in its insufficient theoretical analyses and experiment validation. Specifically:
1. The proposed method is overall an extension to the 1-MLE method in the OSAN, which is not a big problem to me. However, what I would expect is a more in-depth analysis of the proposed method. The authors only use a single category of graphs (CSL and its $(n, l)$ extension) to show that the proposed method is more powerful than a random policy and OSAN, which is trivial from my perspective. Instead of comparing it with random policy, a more interesting question would be: How well does the proposed method compare to the full-bag version? Can it achieve the same expressive power as the full-bag version and in how much $T$ can it be from a theoretical perspective?
2. To distinguish non-isomorphic graphs is only the first step towards an expressive GNN. Can the model successfully encode the structure information like counting cycles play a more important role in real-world tasks. Full-bag versions of subgraph GNNs have a much better ability to encode sub-structures [1]. I am wondering can the proposed method maintain its ability to encode sub-structures.
3. Some commonly used synthetic datasets for comparing expressive power are missing (e.g. EXP [2], CSL [3]). This could be part of the answer to weakness 1.
4. Some commonly used synthetic datasets for evaluating the counting power of GNNs are missing [1]. This could be part of the answer to weakness 2.
5. The main contribution of the subgraph sampling is its lower computational cost compared to the full-bag version. However, the authors only show a simple comparison of the inference time using one dataset where the full-bag version is OOM (Table 7). I believe a more comprehensive comparison between the full-bag version and the proposed method is required. What is the time and memory cost of the proposed method compared to the full-bag version in both the train and the test? How does the cost vary if we increase the $T$?

[1] Huang et al., Boosting the cycle counting power of graph neural networks with i$^2$-GNNs, ICLR23.

[2] Abboud et al., The surprising power of graph neural networks with random node initialization. IJCAI21.

[3] Murphy et al., Relational pooling for graph representations. ICML19.

**Questions:**

1. The current method only works for node-based subgraphs. Could the proposed method be generalized to other policies like edge-based [1] or node-tuple-based [2] subgraphs?


[1] Huang et al., Boosting the cycle counting power of graph neural networks with i$^2$-GNNs, ICLR23.

[2] Qian et al., Ordered subgraph aggregation networks, Neurips22.

---

> ### Author Response · Authors · 2023-11-17
> **Official author response to Reviewer ew9j (1/4)**
>
> We are delighted to see the Reviewer recognized the soundness of our paper while appreciating the presentation of our work. At the same time, the Reviewer raised important points we address in the following.
>
> **Q1**: The proposed method is overall an extension to the 1-MLE method in the OSAN, which is not a big problem to me.
>
> **A1**: We thank the Reviewer for their comment, this gives us the opportunity to further clarify why we believe Policy-Learn is significantly different from OSAN. Our work is primarily motivated by scenarios where a subset of subgraphs is sufficient for maximal expressive power, while the I-MLE method in OSAN aims to make higher-order generation policies (where each subgraph is generated from tuples of nodes) practical. Furthermore, our method generates the bag sequentially, compared to their one-shot generation, and we parameterize the selection network using a Subgraph GNN, that is more expressive than the MPNN used in OSAN. Finally, we do not rely on I-MLE but instead use the simpler implementation of a Gumbel-Softmax trick to enable gradient backpropagation through the discrete sampling process. We proved that these choices lead to a framework that can learn subgraph distributions that cannot be expressed by OSAN.
>
> **Q2**:  How well does the proposed method compare to the full-bag version? Can it achieve the same expressive power as the full-bag version and in how much $T$ can it be from a theoretical perspective?
>
> **A2**: We thank the Reviewer for asking this important question. In certain cases, Policy-Learn can achieve the same expressive power of the full-bag, but with a significantly fewer number of subgraphs.
>
> This is the case for CSL graphs, where only one subgraph is sufficient to achieve the same expressive power of the full bag, as we showed in the paper (Figure 1). This result can be extended to other exemplary non-isomorphic WL-indistinguishable pairs, for example, the pair where one graph consists of two triangles connected by an edge and the other of two squares sharing an edge (Figure 1 in Bevilacqua et al., 2022). Additionally, one subgraph is also sufficient to distinguish strongly-regular graphs of different parameters, which are distinguishable by the full-bag but not by 1-WL.
>
> More interestingly, in certain cases only a _careful selection_ of subgraphs can lead to the same expressive power of the full bag, and our method can provably implement these selections. An example of this case is the family of (n, $\ell$)-CSL graphs where only $\ell$ subgraphs are required to attain the expressive power of the full-bag version, which instead employs $n \cdot \ell$ subgraphs. This example can further be generalized to other families of graphs, obtained from any collection of WL-indistinguishable graphs that become distinguishable when marking any node, e.g., strongly regular graphs of different parameters. Each graph in these families is created by considering $\ell$ disconnected, non-isomorphic instances in the corresponding collection.
>
> **Q3**:  Can the proposed method maintain its ability to encode sub-structures?
>
> **A3**: The Reviewer is raising an interesting point of discussion. The ability of Subgraph GNNs to count substructures can be attributed to the marking of the root nodes, which breaks the anonymity of the marked nodes and allows counting around each marked node (Huang et al., 2023). Interestingly, however, it is possible to show that it is not necessary for a node to be marked in order to count substructures around it: it is sufficient instead that its embedding is unique in its neighborhood. This might happen not only when the node is marked, but also when it receives marking information in the message passing, coming from a root node, and this information is still unique in its surroundings. This implies that our method can exhibit the same counting power of the full-bag whenever each node has an embedding that is unique in its neighborhood, due to the marking information coming from a few selected subgraphs. Note, however, that this might not always be the case, depending on the structure of the graph. For example, it is possible that certain parts of the graph do not see any marking information if we select a small number of subgraphs. This might happen to nodes that are distant from the marked roots of the selected subgraphs, around which a Subgraph GNN necessarily behaves as an MPNN.
>
> We believe that this marks a first step towards the understanding of the counting ability of Subgraph GNNs that rely on a smaller number of subgraphs. However, properly enquiring about this aspect and designing Subgraph GNNs with counting guarantees when considering a subset of subgraphs fall outside the scope of the present work.

---

> ### Author Response · Authors · 2023-11-17
> **Official author response to Reviewer ew9j (2/4)**
>
> **Q4**: Some commonly used synthetic datasets for comparing expressive power are missing (e.g. EXP [2], CSL [3])
>
> **A4**: We welcome the Reviewer's suggestion and additionally experimented with the CSL and the EXP datasets, as shown in the following.
>
>
> | Method                         |             CSL (ACC)            |             EXP (ACC)           |
> |--------------------------------|:----------------------------------:|:-----------------------------------:|
> |   GIN                             |          10.00 +- 0.0              |            51.2 +- 2.1              |
> |                                      |                                           |                                            |
> |   FULL                          |        100.00 +- 0.0              |         100.00 +- 0.0             |
> |   RANDOM $T = 2$      |        100.00 +- 0.0              |           89.92 +- 2.5             |
> |   Policy-Learn $T = 2$  |        100.00 +- 0.0              |         100.00 +- 0.0             |
>
> On the CSL dataset, both RANDOM and Policy-Learn achieve 100% test accuracy, aligning with our theoretical analysis at the beginning of Section 4, where we discussed the disambiguation of these graphs.
>
> On the EXP dataset, Policy-Learn surpasses the performance of RANDOM, and achieves the same performance of the full-bag approach. This numerically confirms the importance of learning which subgraphs to select, rather than randomly sampling them.
>
> **Q5**: Some commonly used synthetic datasets for evaluating the counting power of GNNs are missing.
>
> **A5**: Following the Reviewer’s suggestion, we have additionally tested Policy-Learn on synthetic datasets that measure the ability of models to count substructures. We chose the dataset in Zhengdao et al., 2020, on which full-bag Subgraph GNNs have been extensively evaluated (Zhao et al., 2022, Frasca et al., 2022). We used the dataset splits and evaluation procedure of previous work and reported the average across the seeds in the Table below.
>
>
>
>
> | Method                         |       Triangle       |         Tailed Tri.    |         Star       |        4-Cycle          |
> |--------------------------------|:---------------------:|:---------------------:|:------------------:|:------------------------:|
> |   GIN                             |       0.3569        |         0.2373         |        0.0224    |       0.2185             |
> |                                      |                          |                             |                       |                               |
> |   FULL                           |      0.0183         |        0.0170         |         0.0106   |        0.0210            |
> |                                      |                          |                             |                       |                               |
> |   RANDOM $T = 5$      |       0.1841        |         0.1259         |        0.0105    |        0.1180            |
> |   Policy-Learn $T = 5$  |       0.1658        |         0.1069         |        0.0100    |        0.0996            |
> |                                      |                          |                             |                       |                               |
> |   RANDOM $T = 8$      |       0.1526        |         0.0999         |        0.0119    |        0.0933            |
> |   Policy-Learn $T = 8$  |       0.1349        |         0.0801         |        0.0100    |        0.0793            |
>
>
> While Policy-Learn cannot achieve the same performance of the full-bag, it _significantly_ surpasses the random baseline. This supports our previous discussion on the counting power of our method: although counting all substructures might not be possible with a significantly reduced set of subgraphs, there exist subgraph selections that enable more effective countings. Furthermore, increasing the number of subgraphs proves advantageous in terms of the counting power.

---

> ### Author Response · Authors · 2023-11-17
> **Official author response to Reviewer ew9j (3/4)**
>
> **Q6**: What is the time and memory cost of the proposed method compared to the full-bag version in both the train and the test? How does the cost vary if we increase the T?
>
> **A6**: We believe that to adequately answer this question one should look at the time and space complexity of these methods, which we discuss in the following and we will include in the next manuscript revision. In short, our method scales linearly with $T$. When small values of $T$ are used, as in our paper, this results in a speedup by a factor of $n$ compared to the full-bag Subgraph GNN.
>
> We analyze an equivalent efficient implementation of our method, which differs from Algorithm 1 in that it does not feed to the selection network the entire current bag at every iteration. Instead, it processes only the last subgraph added to the bag, and re-uses the representations of the previously-selected subgraphs, passed though the selection network in previous iterations and stored. We denote $\Delta_\text{max}$ the maximum node degree of the input graph.
>
> **Time Complexity.** The forward-pass asymptotic time complexity of the selection network $f$ amounts to $\mathcal{O}(T \cdot n \cdot \Delta_\text{max})$. This arises from performing $T$ iterations, where each iteration involves selecting a new subgraph by passing the last subgraph added to the bag through the MPNN, which has time complexity $\mathcal{O}(n \cdot \Delta_\text{max})$ and re-using stored representations of previously-selected subgraphs. Similarly, the forward-pass asymptotic time complexity of the prediction network $g$ is $\mathcal{O}((T+1) \cdot n \cdot \Delta_\text{max})$, as each of the $T+1$ subgraphs is processed through the MPNN. Note that, differently from the selection network, the prediction network considers $T+1$ subgraphs as it also utilizes the last selected subgraph.
> Therefore, Policy-Learn has an overall time complexity of $\mathcal{O}((T + (T+1)) \cdot n \cdot \Delta_\text{max})$, i.e., $\mathcal{O}(T \cdot n \cdot \Delta_\text{max})$. The time complexity of full-bag node-based Subgraph GNNs amounts instead to $\mathcal{O}(n^2 \cdot \Delta_\text{max})$, as there are $n$ subgraphs in total.
>
> **Space complexity.** The forward-pass asymptotic space complexity of the selection network $f$ is $\mathcal{O}(T \cdot n + (n +  n \cdot \Delta_\text{max}))$, because we need to store $n$ node features for each of the $T$ subgraphs, and, at each iteration, the space complexity of the MPNN on the subgraph being processed is $\mathcal{O}(n +  n \cdot \Delta_\text{max})$ to store its node features and connectivity in memory.
> Similarly, the forward-pass asymptotic space complexity of the prediction network $g$ is $\mathcal{O}((T+1) \cdot (n +  n \cdot \Delta_\text{max}))$. Therefore, Policy-Learn has an overall space complexity of $\mathcal{O}(T \cdot (n +  n \cdot \Delta_\text{max}))$. The space complexity of the full-bag approach instead amounts to $\mathcal{O}(n \cdot (n +  n \cdot \Delta_\text{max}))$, since the number of subgraphs in the full bag is exactly $n$.
>
> To grasp the impact of this reduction, consider the REDDIT-BINARY dataset where the average number of nodes per graph is approximately $n = 429$, while the number of selected subgraphs is $T=2$.  The time and space complexities are drastically reduced as $T \ll n$, and indeed Policy-Learn can be effectively run while the full bag Subgraph GNN is computationally infeasible.
>
> Finally, we additionally report empirical runtimes on the ZINC dataset in the following. For all methods, we estimated the training time on the entire training set as well as the inference time on the entire test set using a batch size of 128 on an NVIDIA RTX A6000 GPU.
>
> | Method  |  ZINC (Train time (ms))  |  ZINC (Test time (ms)) |  ZINC (MAE) |
> |--|:--:|:--:|:--:|
> |   GIN                             |     1370.10 +- 10.79      |          84.81 +- 0.26         |    0.163 +- 0.004  |
> |                                      |                                      |                                         |                             |
> |   OSAN   $T = 2$          |      2964.46 +- 30.36    |           227.93 +- 0.21      |  0.177 +- 0.016    |
> |                            |                                      |                                         |                             |
> |   FULL                          |      4872.79 +- 14.30     |          197.38 +- 0.30        |  0.087 +- 0.003    |
> |   RANDOM $T = 2$      |     2114.00 +- 27.88      |         107.02 +- 0.22        |  0.136 +- 0.005    |
> |   Policy-Learn $T = 2$  |     2489.25 +- 9.42       |         150.38 +- 0.33         |  0.120 +- 0.003    |
>
> During training, the runtime of Policy-Learn is significantly closer to that of the RANDOM approach than to the one of the full-bag Subgraph GNN FULL, and Policy-Learn is faster than OSAN.
>
> At inference, Policy-Learn places in between RANDOM and FULL, and it is significantly faster than OSAN while also achieving better results.

---

> ### Author Response · Authors · 2023-11-17
> **Official author response to Reviewer ew9j (4/4)**
>
> **Q7**: Could the proposed method be generalized to other policies like edge-based [1] or node-tuple-based [2] subgraphs?
>
> **A7**:  Yes, we can extend our approach to these cases. As for edge-based policies, we can employ the same architecture and output a probability distribution of the edges at the last layer, by using any GNN layer coupled with a final predictor to output edge features. Regarding node $k$-tuples, one option would be to output a distribution on the space of $k$-tuples. However, for $k>2$ this may become impractical in large graphs. Alternatively, nodes in each tuple can be sampled sequentially based on a probability distribution over the nodes, conditioned on previous selections. This strategy can be easily implemented within our framework and it is scalable to large $k$ values even on large graphs.
>
>
>
>
> **References**
>
> Bevilacqua et al., 2022. Equivariant Subgraph Aggregation Networks. ICLR 2022
>
> Huang et al., 2023. Boosting the cycle counting power of graph neural networks with I$^2$-GNNs, ICLR 2023
>
> Zhengdao et al., 2020. Can Graph Neural Networks Count Substructures? NeurIPS 2020
>
> Zhao et al, 2022. From Stars to Subgraphs: Uplifting Any GNN with Local Structure Awareness. ICLR 2022
>
> Frasca et al., 2022. Understanding and Extending Subgraph GNNs by Rethinking Their Symmetries. NeurIPS 2022

---

> > ### Comment · Reviewer_ew9j · 2023-11-17
> > **Thanks for the replies.**
> >
> > I sincerely thank the authors for their detailed replies to all my concerns. After carefully reading the reviews and the response from all reviewers including mine, I think the authors have addressed most of my concerns and I am happy to increase my score towards acceptance.
> >
> > Meanwhile, I have some additional comments:
> >
> > **Just a clarification for Q1.** When I say the proposed method is an extension to 1-MLE in OSAN, I mean an extension in terms of the selection principle and power. It is easy to see that 1-MLE is equivalent to random sampling when the graph is 1-WL indistinguishable (as pointed out in A2 to reviewer VjjR).  The main reason lies in its based encoder is just 1-WL powerful and all subgraphs are sampled at the same time. Therefore, it is a natural thought to extend it by trying iterative sampling, which is the main reason why the proposed method is more powerful than 1-WL.
> >
> > This also brings out the main point from my perspective: designing a sampling method that is more powerful than random sampling is relatively easy. However, how to design a sampling method that can work optimally or nearly optimally in terms of distinguishing ability to all graph classes is hard. Such a property requires a much deeper theoretical analysis. That would be my expectation for a great paper in this line of research.
> >
> > From the theoretical perspective, I would encourage the author to explore the expressive power of the proposed method on more general graph classes in the future, like planar graphs [1]. From the implementation perspective, I am glad to see the discussion from authors about the generalization to more common cases like edge-based or k-node-tuple and would encourage authors to explore it in the future. Meanwhile, there is a very recent work that has a similar motivation and designed a method that is generalizable to the k-node-tuple case [2]. I think the authors could further check and discuss it in the future version.
> >
> > **One minor question**: Is the reported time comparison a result of your newly discussed implementation or the original one?
> >
> > References:
> >
> > [1] Dimitrov et al., PlanE: Representation Learning over Planar Graphs, NeurIPS 2023.
> >
> > [2] Kong et al., MAG-GNN: Reinforcement Learning Boosted Graph Neural Network, NeurIPS 2023.

---

> > > ### Author Response · Authors · 2023-11-17
> > > **Official author response to Reviewer ew9j**
> > >
> > > We are grateful to the Reviewer for the time spent reading our comments, and deeply appreciate the suggestions for future work. We believe it can be an impactful direction for future work to understand the distinguishability power on planar graphs of Subgraph GNNs employing a reduced set of subgraphs, as well as the impact of node tuples. We are further grateful for the references to these contemporary works.
> > >
> > >
> > > The reported time comparison shows the time of the original implementation, which does not re-use the representations of previously-selected subgraphs. Interestingly, this means that it is possible to further speed-up our method with the equivalent efficient implementation.

---

### Official Review · Reviewer_w3Fw · 2023-10-29

**Soundness:** 2 fair
**Presentation:** 3 good
**Contribution:** 3 good
**Rating:** 6
**Confidence:** 4

**Summary:**

The paper focuses on learning effective subgraph selection policies for subgraph GNNs. In particular, it is inspired by an observation that only a small number of subgraphs are needed to differentiate a family of non-isomorphic graphs called the CSL graph. Based on the observation, it constructs a learning-based subgraph selection policy and surpasses previous works on various benchmarks.

**Strengths:**

Generally speaking, I like the efforts on subgraph sampling since the efficiency of subgraph GNNs limits them from being applied in real-world scenarios. In addition, the work is generally motivated and well-written.

**Weaknesses:**

1. Most subgraph sampling strategies face a problem: they cannot guarantee permutation invariance, i.e., generate the same representation for the same graph no matter how the graph is permuted. It seems that the proposed method also cannot guarantee such property as well.

2. I appreciate the efforts in distinguishing the CSL graphs. However, CSL graphs are just a family of regular graphs that cannot be differentiated by 1-WL. Have you analyzed some other families, for example, the strongly regular graphs proposed in (Bodnar et al, 2021b) or some pairs of graphs that are mentioned in (Wang and Zhang 2023)?

3. It seems that the time evaluation is only provided in Table 7, where the time of the full subgraph GNN is not provided due to OOM. I recommend adding time evaluation on more datasets, and reporting the time of "policy learn", "random selection", the full subgraph GNN, and MPNNs. For example, you can report the time of GIN, OSAN, FULL, RANDOM, and POLICY-LEARN on ZINC.

(Wang and Zhang 2023) Wang Y, Zhang M. Towards Better Evaluation of GNN Expressiveness with BREC Dataset. arXiv, 2023.

**Questions:**

Definition 2 and Theorem 1 could possibly lead to some misunderstandings. For example, from my perspective, if the multiset $\\{k_i| i\in \\{1,\dots, l \\}\\}$ is the same, then $CSL(n,(k_1, …, k_l))$ should be isomorphic to each other. The observation should be pointed out, since the definition now seems that the sequence of k_i might also lead to non-isomorphism. In addition, the fact that $CSL (n, k)$ is isomorphic to $CSL (n, n-k)$ would also influence the isomorphism between graphs, which also need to be mentioned.

---

> ### Author Response · Authors · 2023-11-17
> **Official author response to Reviewer w3Fw**
>
> We are thankful to the Reviewer for their constructive comments. We are pleased to notice they have found our work well-motivated and well-written. We proceed by addressing the comments in the following.
>
> **Q1**: Most subgraph sampling strategies face a problem: they cannot guarantee permutation invariance [..]. It seems that the proposed method also cannot guarantee such property as well.
>
> **A1**:  Our architecture is invariant as a function. However, the use of non-deterministic sampling makes the input to the network stochastic and this may result in different representations for the same graph, similar to other approaches that involve randomness in the GNN computation. In practice, however, the non-deterministic nature of the sampling does not seem to hurt the performance, as can be seen in the experimental section.
>
> **Q2**: I appreciate the efforts in distinguishing the CSL graphs [..]. Have you analyzed some other families, for example, the strongly regular graphs proposed in (Bodnar et al, 2021b) or some pairs of graphs that are mentioned in (Wang and Zhang 2023)?
>
> **A2**: Our theoretical analysis can indeed be generalized to other families of graphs. Let $n$ and $\ell$ be natural numbers, and denote by $\mathcal{G}_n$ any set of non-isomorphic yet 1-WL indistinguishable graphs with $n$ nodes that are distinguishable when marking any single node. For example, $\mathcal{G}_n$ can be the set of strongly-regular graphs of different parameters, as suggested by the Reviewer. We define (n, $\ell$)-$\mathcal{G}_n$ to be the family of non-isomorphic graphs, where each graph is obtained by $\ell$ disconnected non-isomorphic instances of $\mathcal{G}_n$. Then, all our theorems are still valid for (n, $\ell$)-$\mathcal{G}_n$. We will discuss these additional families in the next version of our manuscript.
>
> **Q3**: I recommend adding time evaluation on more datasets, and reporting the time of "policy learn", "random selection", the full subgraph GNN, and MPNNs. For example, you can report the time of GIN, OSAN, FULL, RANDOM, and POLICY-LEARN on ZINC.
>
> **A3**: We welcome the Reviewer’s suggestion and we report the runtime of these methods on the ZINC dataset in the following. For all methods, we estimated the inference time on the entire test set using a batch size of 128 on an NVIDIA RTX A6000 GPU.
>
>
> | Method                         |      ZINC (Time (ms))          |        ZINC (MAE)            |
> |--------------------------------|:-----------------------------------:|:-------------------------------:|
> |   GIN                             |      84.81 +- 0.26                |       0.163 +- 0.004         |
> |                                      |                                           |                                       |
> |   OSAN   $T = 2$          |       227.93 +- 0.21             |      0.177 +- 0.016          |
> |                                      |                                           |                                       |
> |   FULL                          |        197.38 +- 0.30            |      0.087 +- 0.003          |
> |   RANDOM $T = 2$      |       107.02 +- 0.22            |      0.136 +- 0.005           |
> |   Policy-Learn $T = 2$  |       150.38 +- 0.33            |      0.120 +- 0.003           |
>
>
> Notably, Policy-Learn is significantly faster than OSAN, while also obtaining better results. Importantly, Policy-Learn places in between the RANDOM approach and the full-bag Subgraph GNN FULL. Furthermore, Policy-Learn takes around 2x the time of the corresponding GIN baseline, but significantly outperforms it.
>
> **Q4**: Definition 2 and Theorem 1 could possibly lead to some misunderstandings. For example, from my perspective, if the multiset $\\{ k_i \vert i \in \\{1, \ldots , \ell \\} \\}$ is the same, then CSL(n, ($k_1, \ldots, k_{\ell}$)) should be isomorphic to each other. The observation should be pointed out, since the definition now seems that the sequence of $k_i$ might also lead to non-isomorphism.
>
> **A4**: The Reviewer is right in their understanding, and for this reason in the paper we always remark that we consider the family of *non-isomorphic* (n, $\ell$)-CSL graphs, where these cases are excluded. However, we will follow the Reviewer’s recommendation and in the next revision of the manuscript we will make this point clearer.

---

> > ### Comment · Reviewer_w3Fw · 2023-11-17
> > **Thanks for your response**
> >
> > I sincerely thank the authors for their response, which has largely addressed my initial concerns.
> >
> > After a meticulous examination of the feedback provided by other reviewers, I have an additional comment: the proposed learning-based sampling strategy, while significantly outperforming the random sampling strategy, is not optimal enough. Take the CSL family as an example, it would be more effective to sample a subgraph from each connected component and employ these samples to distinguish between non-isomorphic graphs. The strategy can also be applied to many other families of regular graphs, where the nodes are treated as the same. Therefore, a potential future work could involve predefining a set of subgraphs at the initial stage, with the aim of reducing the number of steps required in the learning process. This approach could potentially streamline the strategy and enhance its efficiency.

---

> > > ### Author Response · Authors · 2023-11-17
> > > **Official author response to Reviewer w3Fw**
> > >
> > > We would like to express our gratitude to the Reviewer for sharing their insightful comment. We agree that other initial stages are also possible, and we are happy to explore it in future work.

---

### Official Review · Reviewer_gwbR · 2023-10-31

**Soundness:** 3 good
**Presentation:** 3 good
**Contribution:** 2 fair
**Rating:** 8
**Confidence:** 4

**Summary:**

**TLDR**: The paper proposes a learnable subsampling policy to reduce the number of sampled subgraphs in subgraph GNNs.

Subgraph GNNs refer to the family of message-passing graph neural networks which sample subgraphs to improve their expressive power. The paper introduces a new subsampling strategy, _Policy-Learn_ which aims to reduce the number of subgraphs needed for subgraph GNNs. _Policy-Learn_ consists of two subgraph GNNs, one selection network $f$ and one policy network $g$: $f$ learns a distribution over the nodes of the input graph to select which subgraph to sample next, $g$ takes the sampled graphs as input and performs a prediction task. The paper motivates _Policy-Learn_ by considering the graph class of $(n,\ell)$-CSL graphs, where one instance consists of $\ell$ disconnected, non-isomorphic CSL graphs on $n$ nodes. For $(n, \ell)$-CSL graphs, _Policy-Learn_ only needs to sample $\ell$ subgraphs in comparison to random subsampling and the existing subsampling strategy OSAN. In an experimental evaluation, _Policy-Learn_ outperforms OSAN and is, on average, competitive with the presented baseline methods.

**Strengths:**

* The proposed subsampling strategy _Policy-Learn_ is well-motivated and novel.
* _Policy-Learn_ can provably sample subgraphs such that non-isomorphic instances can be distinguished for one specific graph class ($(n, \ell)$-CSL graphs) on which 1-WL fails.
* In the experimental evaluation, _Policy-Learn_ is competitive with most baseline methods and outperforms OSAN

**Weaknesses:**

* Theoretical limitations: While the presented theoretical results are novel and interesting, they also appear to be limited. The artificially constructed graph class $(n, \ell$)-CSL is 1-WL indistinguishable; however, higher-order models and GNN variants are able to distinguish them. A more comprehensive analysis of the expressive power of _Policy-Learn_ could strengthen the contribution significantly.
* Clarity: Although the paper is generally well-written, more precise language would improve readability:
     * "[...] preventing the applicability of Subgraph GNNs on important datasets" -> What datasets are important?
    * "Contributions: [...] An experimental evaluation of the new approach demonstrating its advantages." -> It would be more informative if the advantages are specified.
    * " [...] which includes feature aggregation in light of the alignment of nodes [...]" -> Could you specify what that means
    * "[...] and demonstrate that our framework performs better on real-word datasets" -> Better than what?

**Questions:**

1. **Expressiveness**: _Policy-Learn_ can distinguish non-isomorphic instances in the graph class $(n, \ell)$-CSL, which are indistinguishable by 1-WL. What about the opposite? Can you characterize graph classes whose non-isomorphic instances are provably indistinguishable by _Policy-Learn_? Are there graph classes where OSAN is stronger than _Policy-Learn_? Is _Policy-Learn_ limited by higher-order WL?
2. **Assumption in proof of Theorem 4**: Is the (necessary) assumption that $f$ has $n$ layers feasible for larger graphs? Do you have experimental results on $(n, \ell)$-CSL graphs?
3. **Extension of theoretical results**: Have you thought about extensions of your theoretical results, e.g., other graph classes where marking any node in a graph is sufficient or where marking a limited number of nodes is sufficient?
4. **Experiments**:

    a. How did you choose the values for $T$ (2 and 5 in Tables 1 and 3, 2 and 20 in Table 2)?

    b. In Section 7, paragraph ZINC: "Notably, OSAN performs worse than our random baseline due to differences in the implementation of the prediction network". Could you elaborate on the differences and why this affects the performance?
5. **Time comparison**: How does _Policy-Learn_ compare with respect to time vs. prediction performance in comparison to more expressive GNNs (e.g., GSN, CIN)?

---

> ### Author Response · Authors · 2023-11-17
> **Official author response to Reviewer gwbR (1/2)**
>
> We appreciate the fact that the Reviewer highlighted the novelty of our method and our theoretical results in their feedback. At the same time, the Reviewer raised important points we address in the following.
>
> **Q1**:  Clarity: Although the paper is generally well-written, more precise language would improve readability.
>
> **A1**: We thank the Reviewer for their suggestions, and we will make sure to include them in the next revision of the paper.
>
> **Q2**:  Expressiveness: Can you characterize graph classes whose non-isomorphic instances are provably indistinguishable by Policy-Learn? Are there graph classes where OSAN is stronger than Policy-Learn? Is Policy-Learn limited by higher-order WL?
>
> **A2**: This is a very interesting aspect and we thank the Reviewer for bringing it up.
>
> Interestingly, we can now prove that Policy-Learn can distinguish any strongly regular graphs of different parameters, but, just like 3-WL, cannot distinguish any strongly regular graphs of the same parameters. This implies that Policy-Learn is not as powerful as 4-WL.
>
> Importantly, there are no graph classes where OSAN is stronger than Policy-Learn. Specifically, for any instantiation of OSAN, there exists a set of weights for Policy-Learn such that it outputs exactly the same probability distribution. In other words, Policy-Learn can parameterize all the probability distributions that OSAN can. The contrary, however, is not true: there exist probability distributions that Policy-Learn can parameterize but OSAN cannot.
>
> We will expand this expressivity analysis in the next revision of the manuscript.
>
> **Q3**: Assumption in proof of Theorem 4: Is the (necessary) assumption that f has n layers feasible for larger graphs?
>
> **A3**: We thank the Reviewer for bringing this point up to our attention. First, we would like to remark that the number of nodes in a (n,$\ell$)-CSL graph is $n \cdot \ell$, which is much larger than $n$, the value we used as a loose bound on the number of layers in the proof. However, we understand the Reviewer’s concern, and we have therefore now proved that the bound can be tightened to $n/2$ by considering the circular structure of the connected components. We believe that this bound can be further reduced by taking into account the skip connections.
>
> **Q4**: Extension of theoretical results: Have you thought about extensions of your theoretical results, e.g., other graph classes where marking any node in a graph is sufficient or where marking a limited number of nodes is sufficient?
>
> **A4**: In the paper, we showed that for CSL graphs it is sufficient to mark any single node for distinguishability (Figure 1). However, this result can be extended to other examples of  non-isomorphic WL-indistinguishable pairs, for example, the pair where one graph consists of two triangles connected by an edge and the other of two squares sharing an edge (Figure 1 in Bevilacqua et al., 2022). Additionally, marking any node is also sufficient to distinguish strongly-regular graphs of different parameters.
>
> Importantly, it is also possible to consider other families of graphs beyond the (n, $\ell$)-CSL graph family where marking a limited number of nodes is sufficient for disambiguation. Indeed, our theoretical results are valid for any family of graphs, as we describe next, obtained from any collection of WL-indistinguishable graphs that become distinguishable when marking any node, e.g., strongly regular graphs of different parameters. Similarly to the construction of the (n, $\ell$)-CSL family in our paper, each graph in these families is created by considering $\ell$ disconnected, non-isomorphic instances in the corresponding collection. Finally, we believe that it is possible to extend this analysis to certain families of connected graphs, and we leave this aspect for future work.

---

> ### Author Response · Authors · 2023-11-17
> **Official author response to Reviewer gwbR (2/2)**
>
> **Q5**: Experiments: How did you choose the values for $T$ (2 and 5 in Tables 1 and 3, 2 and 20 in Table 2)?
>
> **A5**: Experimentally, our goal was to show that it is possible to obtain compelling results on standard benchmarks even when considering a significantly smaller number of subgraphs than the total typically employed by existing methods. Therefore we chose $T=2$ and $T=5$ as two exemplary numbers, and demonstrated that even with only $T=2$ subgraphs Policy-Learn achieves very good performance. Nonetheless, we have further experimented with other values of $T$, namely $T=3$ and $T=8$, which we report in the following.
>
>
> | Method                       |          ZINC (MAE)             |
> |------------------------------|:-----------------------------------:|
> |   GIN                           |        0.163 +- 0.004            |
> |                                    |                                            |
> |   FULL                        |         0.087 +- 0.003           |
> |   RANDOM $T = 3$    |        0.128 +- 0.004           |
> |   Policy-Learn $T = 3$|         0.116 +- 0.008          |
> |   RANDOM $T = 8$    |         0.102  +- 0.003         |
> |   Policy-Learn $T = 8$|         0.097 +- 0.005          |
>
> These results align with the observations made for other values of $T$. In particular, Policy-Learn always outperforms the random baseline, and the gap is larger when the number of subgraphs $T$ is smaller, where selecting the most informative subgraphs is more crucial.
>
> **Q6**: Experiments: In Section 7, paragraph ZINC: "Notably, OSAN performs worse than our random baseline due to differences in the implementation of the prediction network". Could you elaborate on the differences and why this affects the performance?
>
> **A6**:  We ran our random baseline on exactly the same prediction network that we used for Policy-Learn to ensure a direct comparison. This comprises architectural choices that are different from those taken in OSAN, including the number of layers, the embedding dimension, and the implementation of the residual connections. In our next revision, we will make this point clearer and further include the performance of the RANDOM baseline using the same prediction network considered in OSAN, as reported by Qian et al., 2022.
>
> **Q7**: Time comparison: How does Policy-Learn compare with respect to time vs. prediction performance in comparison to more expressive GNNs (e.g., GSN, CIN)?
>
> **A7**: We welcome the Reviewer’s suggestion, and we report the time and the prediction performance on the ZINC dataset in the following. For all methods, we report the inference time on the entire test set using a batch size of 128. The runtime of CIN is taken from the original paper, and it is measured on an NVIDIA Tesla V100 GPU. To ensure a fair comparison, we therefore timed Policy-Learn on the same GPU type.
>
>
> | Method                         |      ZINC (Time (ms))          |        ZINC (MAE)            |
> |--------------------------------|:----------------------------------:|:--------------------------------:|
> |   GIN                             |        126.91 +- 0.82            |       0.163 +- 0.004         |
> |   CIN                             |          471.00 +- 3.00          |       0.079 +- 0.006         |
> |   Policy-Learn $T = 2$  |          235.14 +- 0.21         |      0.120 +- 0.003           |
> |   Policy-Learn $T = 5$  |           411.19 +- 0.39         |      0.109 +- 0.005          |
>
> First, we observe that Policy-Learn is faster than CIN, especially considering the additional preprocessing time required by CIN for the graph lifting procedures, which is not measured in the Table.
>
> Second, although CIN outperforms Policy-Learn, it should be noted that CIN explicitly models cycles and rings, which are obtained in the preprocessing step and serve as a domain-specific inductive bias. On the contrary, the subgraph generation policy in Policy-Learn (i.e., node-marking) is entirely domain-agnostic and not tailored to the specific application.
>
> Finally, even though we did not find the runtime of GSN in the original paper, it is likely to be similar to that of GIN as GSN performs message passing on node features augmented with substructure counts. However, similarly to CIN, GSN requires an additional preprocessing time for the substructure counting that should be taken into account and also requires additional domain knowledge to choose which substructures to count.
>
> **References**
>
> Bevilacqua et al., 2022. Equivariant Subgraph Aggregation Networks. ICLR 2022

---

> ### Comment · Reviewer_gwbR · 2023-11-20
>
> Thank you for the detailed response about the expressive power of Policy-Learn and the additional experiments, I am looking forward to the revised submission!

---

> > ### Author Response · Authors · 2023-11-20
> > **Official author response to Reviewer gwbR**
> >
> > We are grateful to the Reviewer for their comments and we have now uploaded a revised submission, which includes the suggested improvements. As recommended by the Reviewer, we have additionally included: clarifications and extensions of our theoretical results in Appendix D, additional experiments for other values of $T$ in Appendix E, a time comparison featuring the comparison with CIN in Appendix E.

---

> ### Comment · Reviewer_gwbR · 2023-11-23
>
> Thanks for uploading the revised manuscript. As you addressed all of my concerns, I raised my score.
>
> Please note that you have typos in p. 19-20:
> - "... graph Definition 2, where marking a limited number of nodes is sufficient for identifiability ..."
> - "Let G be a strongly regular graphs ..."
> - "... can be parameterize exactly ..."

---

### Official Review · Reviewer_VjjR · 2023-11-01

**Soundness:** 3 good
**Presentation:** 2 fair
**Contribution:** 2 fair
**Rating:** 6
**Confidence:** 3

**Summary:**

Subgraph GNNs generally refers to GNN methods that run GNNs on several subgraphs obtained from the input graph. Recently, a variety of such methods, differing from each other in the subgraph selection policies, are proposed. Of particular relevance to the current paper is the OSAN framework by Qian et al, in which the subgraph selection policy is learned. In the current paper, a more expressive GNN architecture DS-GNN is used (rather than a classical GNNs) of the subgraph selection policy. This DS-GNN provides a distribution from which is sampled using standard Gumbel softmax trick. It is shown theoretically the approach can be more powerful than the OSAN approach.

**Strengths:**

1. Research in how to learn subgraph selection policies is highly relevant in view of the popularity of subgraph GNN approaches.

2. Related work is well described.

3. The policy learn method is well designed.

4. Theoretical guarantees over special classes of CSL graphs are presented. In particular, it is argued that the proposed approach can be stronger than a previous approach.

**Weaknesses:**

1. It is not clearly described what gives the proposed method more power than e.g., OSAN.

2. The method seems to depend on a subgraph GNN method (DS-GNN) which high computational cost.

**Questions:**

**Q1** Could you explain the histograms in Figure 1 after labeling one vertex?

**Q2** What is the ingredient of the method which results in more power than say OSAN?

**Q3** Is the proposed method at least as powerful as OSAN or incomparable? What about comparisons with other subgraph formalisms?

**Q4** A number of subgraphs are selected in order to reduce complexity. However, the DS-GNN method used for selection policy relies on all subgraphs? What is the overall complexity of the method?

---

> ### Author Response · Authors · 2023-11-17
> **Official author response to Reviewer VjjR (1/2)**
>
> We are glad to see that the Reviewer has appreciated the relevance of the research topic while finding our proposed method well-designed and the related work well-described. They have nonetheless raised a few questions that we address below.
>
> **Q1**: Could you explain the histograms in Figure 1 after labeling one vertex?
>
> **A1**: For each original graph, we consider the application of the WL test to the subgraph obtained by marking the crossed node. More precisely, at time step 0, each non-marked node in the subgraph is assigned the same initial constant color, which differs from the initial color of the marked node. Without loss of generality, denote these colors as color 1 and color 0, respectively. Then, the WL test proceeds by refining the color of each node by aggregating the colors of its neighbors (including itself). More precisely, the new color of a node is obtained through a hash function taking as input the multiset of colors of the neighbors and of the node itself. It is easy to see that, at time step 1, the marked node has a unique color, which we call  color 2, its neighbors all have the same colors, color 3, and all the remaining nodes have the same color (different from the marked node and its neighbors), color 4. The refinement is repeated until we reach a stable coloring. At this point, we simply collect the number of nodes having the same color in a histogram. For the upper graph in Figure 1, the histogram indicates that in the stable coloring, there are 4 blue nodes, 4 green nodes, 4 yellow nodes, and 1 orange node. Importantly, since the WL test represents a necessary condition for graph isomorphism, different histograms imply that the two graphs are not isomorphic, and thus the WL test distinguishes them. Figure 1 shows that marking only one node is sufficient for distinguishing the two graphs, which are instead indistinguishable when no node is marked, as the WL test returns the same histogram for them (13 orange nodes).
>
> **Q2**: What is the ingredient of the method that results in more power than say OSAN?
>
> **A2**: The iterative procedure that selects one subgraph at a time based on previous selections, paired with the Subgraph GNN used to implement the selection network $f$, are the ingredients that result in more power than OSAN.
>
> Indeed, recall that OSAN selects all subgraphs at the same time, by applying an MPNN over the original graph only. Consider again the family of non-isomorphic (n, $\ell$)-CSL graphs we studied in the paper, which contains WL-indistinguishable graphs composed of disconnected copies of WL-indistinguishable (sub-)graphs. Given that OSAN uses the original graph, then it cannot differentiate between the different CSL (sub-)graphs and will return a uniform probability distribution over all the nodes. This implies that it cannot ensure that the marked nodes will belong to different CSL (sub-)graphs, which represents a necessary condition for identification (Proposition 2). On the contrary, Policy-Learn iteratively adds a new subgraph based on the subgraphs already present in the bag. Whenever a subgraph is in the bag, then all nodes in the CSL (sub-)graph of its marked root node will have a different color than the others. Thus it is always possible to distinguish nodes belonging to CSL (sub-)graphs having a marked node, and therefore the final MLP can assign these nodes a zero probability while maintaining a uniform probability over the remaining nodes. This implies that the next node to be marked will be sampled from a CSL (sub-)graph that has no marked node yet, effectively implementing the efficient policy $\pi$ which is sufficient for identification.
>
> **Q3**: Is the proposed method at least as powerful as OSAN or incomparable? What about comparisons with other subgraph formalisms?
>
> **A3**: The proposed method Policy-Learn is _strictly more powerful_ than OSAN, although the term is more convoluted in this context due to the probabilistic nature of the sampling procedure.
>
> More formally, for any instantiation of OSAN, there exists a set of weights for Policy-Learn such that it outputs exactly the same probability distribution. In other words, Policy-Learn can parameterize all the probability distributions that OSAN can. The contrary, however, is not true: there exist probability distributions that Policy-Learn can parameterize but OSAN cannot. An exemplary case is the one that yields to the policy $\pi$ considered for the family of non-isomorphic (n, $\ell$)-CSL graphs, allowing to sample all nodes from different CSL (sub-)graphs.
>
> More broadly, it is possible to show that Policy-Learn is more expressive than 1-WL, and not as powerful as 4-WL as it cannot distinguish strongly-regular graphs of the same parameters. We will discuss these aspects more thoroughly in the next version of our manuscript.

---

> ### Author Response · Authors · 2023-11-17
> **Official author response to Reviewer VjjR (2/2)**
>
> **Q4**: The DS-GNN method used for selection policy relies on all subgraphs? What is the overall complexity of the method?
>
> **A4**: The DS-GNN method used for the selection policy *does not* rely on all subgraphs. On the contrary, at each timestep $t$, it uses only the current bag, which contains $t$ subgraphs, to select the next subgraph to be added to the bag, starting from the original graph. This means that the number of subgraphs passed through the selection network is significantly smaller than the total number.
>
> Nonetheless, we acknowledge the fact that including the computation complexity of the method would improve the quality of the manuscript. Thus, we present it below and we will include it in the next revision.
>
>
> We analyze an equivalent efficient implementation of our method, which differs from Algorithm 1 in that it does not feed to the selection network the entire current bag at every iteration. Instead, it processes only the last subgraph added to the bag, and re-uses the representations of the previously-selected subgraphs, passed though the selection network in previous iterations and stored.
> In what follows, we consider the feature dimensionality and the number of layers to be constants. We denote $\Delta_\text{max}$ the maximum node degree of the input graph, $n$ the number of its nodes and $T$ the number of selected subgraphs.
>
> The forward-pass asymptotic time complexity of the selection network $f$ amounts to $\mathcal{O}(T \cdot n \cdot \Delta_\text{max})$. This arises from performing $T$ iterations, where each iteration involves selecting a new subgraph by passing the last subgraph added to the bag through the MPNN, which has time complexity $\mathcal{O}(n \cdot \Delta_\text{max})$ and re-using stored representations of previously-selected subgraphs. Similarly, the forward-pass asymptotic time complexity of the prediction network $g$ is $\mathcal{O}((T+1) \cdot n \cdot \Delta_\text{max})$, as each of the $T+1$ subgraphs is processed through the MPNN. Note that, differently from the selection network, the prediction network considers $T+1$ subgraphs as it also utilizes the last selected subgraph.
>
> Therefore, Policy-Learn has an overall time complexity of $\mathcal{O}((T + (T+1)) \cdot n \cdot \Delta_\text{max})$, i.e., $\mathcal{O}(T \cdot n \cdot \Delta_\text{max})$. The time complexity of full-bag node-based Subgraph GNNs amounts instead to $\mathcal{O}(n^2 \cdot \Delta_\text{max})$, as there are $n$ subgraphs in total.
>
> To grasp the impact of this reduction, consider the REDDIT-BINARY dataset, which we have experimented on to make this point clear (Tables 2 and 7). In this dataset the average number of nodes (and thus subgraphs) per graph is approximately $n = 429$, making the full-bag Subgraph GNN infeasible. In our experiment the number of selected subgraphs is $T=2$. The time complexity is drastically reduced as $T \ll n$, and, indeed, Policy-Learn can be effectively and successfully run.

---

> ### Comment · Reviewer_VjjR · 2023-11-18
> **Comments on Authors' answers 1/2**
>
> I would like to thank the authors for their clear answers to my question. For Q1, please also clarify this in the paper. For the remaining questions: When you say that it is not as powerful as 4-WL, do you mean that it is bounded by 4-WL as well?

---

> ### Comment · Reviewer_VjjR · 2023-11-18
> **Comments on Authors' answers 2/2**
>
> Thanks for explaining that your approach does not use all subgraphs for policy learning. Please add the computational complexity part to the paper (or supp material).

---

> > ### Author Response · Authors · 2023-11-20
> > **Official author response to Reviewer VjjR**
> >
> > We are thankful to the Reviewer for their constructive feedback. We have uploaded a new revision which includes in Appendix B the explanation of Figure 1, thus clarifying the answer to Q1. Following the Reviewer’s suggestion, we have further included the complexity analysis in Appendix F.
> >
> > Finally, we would like to clarify that we have proved that Policy-Learn is not as powerful as 4-WL, which means that there exists a pair of 4-WL distinguishable graphs that cannot be distinguished with our method. This, however, does not necessarily imply that Policy-Learn is upper-bounded by 4-WL, as the two might in principle be incomparable. While we conjecture that in reality this is not the case, we believe that properly characterizing the expressive power upper bound of Subgraph GNNs employing a reduced set of subgraphs would require careful analysis, which we are eager to make in a follow-up work.

---

### Author Response · Authors · 2023-11-17
**Official Response**

We would like to express our gratitude to all Reviewers for their valuable feedback and insightful comments.


We are glad to see that our work has been positively received. The Reviewers have found the research problem *“highly relevant”* (**VjjR**) and our method *“well-motivated”* (**w3Fw**, **gwbR**, **VjjR**), *“well-designed”* (**VjjR**) and *“novel”* (**gwbR**). They have also recognized the validity of our theoretical contributions, considered *“novel and interesting”* (**gwbR**) and *“sound”* (**ew9j**). Lastly, we are pleased to notice that all Reviewers have appreciated the presentation of our work, finding the paper *“well-written”* (**w3Fw**, **gwbR**) and *“easy to follow”* (**ew9j**).


**Additional Expressive Power Analysis.** Following the Reviewers’ suggestions, we have expanded our theoretical analysis. We have generalized all our theoretical results to wider families of graphs, therefore extending cases where Policy-Learn is as powerful as the full-bag, and strictly more powerful than OSAN and the random baseline. We have further proved that Policy-Learn is not as powerful as 4-WL, as, like 3-WL, it cannot distinguish strongly regular graphs of the same parameters. Finally, we have discussed conditions related to the counting power of Subgraph GNNs that use a small number of subgraphs.


**New Experiments.** Several additional experiments were conducted following the Reviewers’ comments: (1) Various thorough time comparisons featuring diverse methods, showing our method is not only faster than OSAN and the full bag version, but also more efficient than expressive, domain-specific architectures like CIN; (2) Two expressivity benchmarks, CSL and EXP, clearly demonstrating the advantage of learning which subgraphs to select, rather than randomly sampling them; (3) Counting datasets, aligning with our theoretical discussion on the counting power of Policy-Learn. Details are provided in the individual responses.


We will follow up by uploading a new revision of our manuscript and we are happy to engage in further discussion.

---

### Author Response · Authors · 2023-11-20
**New Manuscript Revision**

We would like to bring to the attention of the Reviewers the new paper revision we have uploaded. The revision includes the changes outlined in the individual responses, and visualized in _blue_.


The changes include:
- The extension of our theoretical analysis to new families of graphs in Appendix D, as recommended by Reviewers **gwbR** and  **w3Fw**;
- A clarification that our approach is more powerful than OSAN and can attain the same expressive power of the full bag in Appendix D, as recommended by Reviewers **VjjR**, **gwbR** and **ew9j**;
- Results from new experiments on CSL, EXP, Counting datasets in Appendix E, as recommended by Reviewer **ew9j**;
- Results from time comparisons in Appendix E, as recommended by Reviewers **gwbR**,  **w3Fw** and **ew9j**;
- Complexity analysis in Appendix F, as recommended by Reviewers **VjjR** and **ew9j**.

---

### Author Response · Authors · 2023-11-22
**Official author comment**

We express our sincere gratitude to all Reviewers for their constructive feedback. The revised manuscript incorporates the suggestions provided, including the additional experiments along with an extension of our theoretical analysis.

If the Reviewers find that we have adequately addressed their concerns, we would be grateful if they would consider reassessing their rating.

We are also open to further discussion and welcome any additional input.

---

### Meta-Review · Area_Chair_fsgq · 2023-12-15

**Metareview:**

The paper addresses the computational complexity associated with message passing on many subgraphs in Subgraph Graph Neural Networks (GNNs), which are expressive architectures for learning graph representations from sets of subgraphs. The focus is on learning to select a small, data-driven subset of potential subgraphs to mitigate this complexity. The authors first motivate the problem by proving the existence of efficient subgraph selection policies for families of WL-indistinguishable graphs, demonstrating the potential for small subsets to identify entire graph families. They introduce a novel approach, Policy-Learn, which iteratively learns how to select subgraphs. Unlike random policies and prior methods, Policy-Learn is proven to learn efficient policies. Experimental results show that Policy-Learn outperforms existing baselines across various datasets.

All reviewers agree that the paper should be accepted. It provides an interesting novel idea, discusses related work well, and shows good empirical performance. The rebuttal has also clarified some of the questions.

**Justification For Why Not Higher Score:**

-

**Justification For Why Not Lower Score:**

-

---

### Decision · Program_Chairs · 2024-01-16

Accept (poster)